# Slc11 Synapomorphy: A Conserved 3D Framework Articulating Carrier Conformation Switch

**DOI:** 10.3390/ijms242015076

**Published:** 2023-10-11

**Authors:** Mathieu F. M. Cellier

**Affiliations:** Centre Armand-Frappier Santé Biotechnologie, Institut National de la Recherche Scientifique (INRS), Laval, QC H7V 1B7, Canada; mathieu.cellier@inrs.ca

**Keywords:** AF2-CF modeling, LeuT fold, Slc11 synapomorphy, carrier conformation switch, phylogenetic analysis, in silico mutagenesis, epistasis, proton-dependent Mn^2+^ transporter (MntH), natural resistance-associated macrophage protein (Nramp), MntH C (MC)

## Abstract

Transmembrane carriers of the Slc11 family catalyze proton (H^+^)-dependent uptake of divalent metal ions (Me^2+^) such as manganese and iron—vital elements coveted during infection. The Slc11 mechanism of high-affinity Me^2+^ cell import is selective and conserved between prokaryotic (MntH) and eukaryotic (Nramp) homologs, though processes coupling the use of the proton motive force to Me^2+^ uptake evolved repeatedly. Adding bacterial piracy of *Nramp* genes spread in distinct environmental niches suggests selective gain of function that may benefit opportunistic pathogens. To better understand Slc11 evolution, Alphafold (AF2)/Colabfold (CF) 3D predictions for bacterial sequences from sister clades of eukaryotic descent (MCb and MCg) were compared using both native and mutant templates. AF2/CF model an array of native MCb intermediates spanning the transition from outwardly open (OO) to inwardly open (IO) carriers. In silico mutagenesis targeting (i) a set of (evolutionarily coupled) sites that may define Slc11 function (putative synapomorphy) and (ii) residues from networked communities evolving during MCb transition indicates that Slc11 synapomorphy primarily instructs a Me^2+^-selective conformation switch which unlocks carrier inner gate and contributes to Me^2+^ binding site occlusion and outer gate locking. Inner gate opening apparently proceeds from interaction between transmembrane helix (h) h5, h8 and h1a. MCg1 xenologs revealed marked differences in carrier shape and plasticity, owing partly to an altered intramolecular H^+^ network. Yet, targeting Slc11 synapomorphy also converted MCg1 IO models to an OO state, apparently mobilizing the same residues to control gates. But MCg1 response to mutagenesis differed, with extensive divergence within this clade correlating with MCb-like modeling properties. Notably, MCg1 divergent epistasis marks the emergence of the genus *Bordetella*-*Achromobacter*. Slc11 synapomorphy localizes to the 3D areas that deviate least among MCb and MCg1 models (either IO or OO) implying that it constitutes a 3D network of residues articulating a Me^2+^-selective carrier conformation switch which is maintained in fast-evolving clades at the cost of divergent epistatic interactions impacting carrier shape and dynamics.

## 1. Introduction

The physiological hallmark of transmembrane carriers of the Slc11 family is to catalyze H^+^-dependent, high-affinity import of Me^2+^ such as manganese (Mn) and ferrous iron (Fe) into the cytoplasm [1,2]. Slc11 carriers use a chemiosmotic mechanism of transport rooted deep in their evolutionary history, the ‘rocking bundle’ (or ‘gated channel’) mechanism [3] that was discovered after solving structures of the LeuT fold [4] and that may be declined in as many versions as diverse families form the superfamily APC (PFAM Clan CL0062) [5]. Among these, PFAM family 1566 comprises both the Slc11 family and phylogenetic outgroups (orphan and NRMT types) [6,7,8], which diverged specifically at the Me^2+^ substrate binding site (Me^2+^ BS) plus several sites that together remained (quasi) invariant in Slc11 carriers. These sites constitute type ii evolutionary rate shifts [9], and their co-occurrence is hereafter referred to as ‘Slc11 synapomorphy’ [10,11]. As such, the Slc11 basic mechanism of high-affinity Me^2+^ cell import is considered family-specific and evolutionarily conserved (Figure 1, left panel).

Yet, detailing the broad phylogeny of Slc11 carriers shows stepwise evolution of a transmembrane H^+^ network that connects residues (polar or charged) within the rather immobile ‘hash module’ of the carrier (vs. the ‘rocking bundle’ and ‘gating parts’) [3,12]. Serial synapomorphies, i.e., sets of co-evolutionary rate shifts exclusively shared among taxa from successive monophyletic clades across Slc11 phylogeny, showed stepwise evolution of the Slc11 H^+^ network, e.g., as (i) the Slc11 family arose and (ii) early bacteria evolved from anaerobiosis to aerobiosis; when eukaryotic cells (iii) emerged and (iv) complexified while dealing with potential microbial invaders. Molecular data thus imply that the Slc11 mechanism coupling the use of the proton motive force to Me^2+^ uptake evolved repeatedly, as novel environmental conditions prevailed (Figure 1, right panel).

Against this backdrop, added complexity comes from independent episodes of bacterial piracy of *Nramp* genes (encoding Slc11 eukaryotypes pN-I and pN-II) subsequently disseminated in distinct environmental niches [12]. This process suggests selective gain of function and opportunity for innovation in various infectious settings [14]. The elemental nature of Slc11 preferred substrates (Mn and Fe) makes them vital resources highly coveted during infection [15], for different reasons perhaps, as Fe is indissociable from the growth of cells able to handle it safely while Mn is primarily required to resist conditions adverse to growth [16,17]. As the competitive import of Fe and/or Mn may confer a survival advantage, including during infection, it likely exerts strong selective pressure on Slc11 carrier efficiency.

Molecular phylogeny indicates that the precise mechanism coupling the use of the proton motive force with Me^2+^ uptake differs between MntH clades of bacterial origin (e.g., MntH B, MntH A) and those derived from eukaryotic ancestors (MCa, MCb and MCg on one hand, and on the other hand, MCaU (Figure 1)). Bacterial *Nramp* descendants (coding for MntH Cs or MCs) could further diversify by adapting to select ecological niches (e.g., the gut of pollinating insects for MCb and plant rhizosphere for MCg). Interestingly, AF2 modeling of MC sequences yielded two sorts of predictions wherein MCb sequence diversity produced an array of plausible conformers while in contrast, models predicted for MCg, MCa and MCaU xenologs appeared rather stereotypical [12].

Arguably, as MCb structures solved previously were used to train AF2 learning models, subsequent modeling may in turn interpret MCb sequence diversity as available space allowing accurate prediction of various conformers, which could indicate MCb carrier intrinsic flexibility. On the other hand, MC sequences from other groups, e.g., MCg, differ from MCb from two standpoints: a unique founding event followed by independent evolutionary trajectories. It thus seems that the complex evolutionary processes at work constrain AF2 modeling space for other MC subgroups, yielding predicted structures that appear relatively rigid. Differences in the intrinsic flexibility of MCb and MCg carriers may reflect divergent adaption of H^+^-dependent Mn^2+^ import dynamics to selective niches. To investigate this possibility, AF2/CF modeling of three MCbs and two MCg1s was subjected to a site-directed mutagenesis scheme designed to either induce or prevent a conformation switch to seek evidence of sister-group-specific structural divergence of possible functional significance.

To demonstrate that MCb AF2/CF models represent functional conformers, structural ensembles capturing alternate carrier states, e.g., either outwardly open to bind substrate, inwardly open for its intracellular delivery or in-between, were collected. Intramolecular communities of networked residues that would typify these ensembles and could be targets for mutagenesis to probe their role in carrier conformation exchange were identified. A mutational strategy aimed at inducing an ‘in silico conformation switch’ that would mimic the transition observed between native conformer ensembles was established. To test the hypothesis that Slc11 synapomorphy may constitute a 3D network of conformation switch instructive sites, mutation combinations that reproduced ancestral evolutionary rate shifts were employed. A suitable genetic background, i.e., mutation combination leading to alternate conformer modeling relative to the native template, was used to question the role of several communities of networked residues. Distinct MCb templates were used to compare results and appreciate shared functional properties. 

Knowledge gained from studying the MCb in silico conformation switch was applied to characterize its sister clade MCg. To establish MCg1 models as functional conformers, the conservation of Slc11 synapomorphy’s role in conformation exchange was tested with distinct templates. It was determined if MCg1 models undergo a similar mutation-induced conformation switch. Template sensitivity to mutation combinations inducing alternate conformer modeling was compared between MCg1s and MCbs and among divergent MCg1s. Structural, phylogenetic and taxonomic evidence of functional significance relating to the divergence between MCg1 and MCb clades, and amongst them, was gathered to support data interpretation. Three-dimensional conservation of conformation switch instructive sites was examined across MCg1 and MCb models. Integrating all the data produced suggests a plausible evolutionary scenario wherein 3D conservation of Slc11 synapomorphy embeds the basic mechanism of carrier conformation exchange, which is maintained at a substantial cost in fast-evolving MC clades through divergent epistasis.

## 2. Results

### 2.1. AF2/CF Predict Native MCb Carrier Cycle Intermediates

Slc11 carriers work by cycling between alternative conformations using the proton motive force to direct Mn^2+^ transport toward the cell interior [1,18]. During Mn^2+^ import, Slc11 carrier conformation evolves through a continuum of conformers from an OO state (e.g., apo, ion-coupled, metal-bound) to an IO state (e.g., metal-bound, ion-coupled, apo) with intermediate conformers (ion-coupled and metal-bound) doubly occluded and including the tipping point for directional transport [5,19]. Notably, AF2/CF predictions for Uniprot MCb sequences produce an array of 3D models that may represent functional intermediates in the forward transition from an OO to an IO state [12] (Figure 2).

To infer a temporal sequence of possible structural changes accompanying MCb conformation transition, the solved pdb entries 5M87 and 5M94 were set as reference structures representing distal alternate 3D states (OO and IO, respectively). MCb AF2/CF models were selected among those populating the shortest path linking MCb reference structures in AAA Dali comparisons, including one that seemed close to the MCb conformation switch point (Figure 2). Per-residue root mean square deviation of the Ca chain (RMSD) from either reference structure was calculated for each candidate conformer selected (Figure 3).

To model the structural evolution of the MCb carrier during OO-to-IO transition, three candidate conformers, picturing hypothetical steps close to the MCb switch point (either OO, IO or in-between), were altogether superposed with the reference structures (Figure 4).

Initial progression through the OO state shows deviation of the pseudo-symmetric segments h1b and h6a (Figure 4, left panel), which increases until conversion to the IO state accompanied by displacement of h9–h10 and movement h6a/h10 may push h1b toward h5 creating a pivot point for h5 tilt (Figure 4, middle panel). Bending of h8 in the 5M94 structure alters the topology of the MCb H^+^ network and could contribute to opening the inner gate via h5 (Figure 4, right panel).

The detailed motion of h1 and h6 (Figure 5), each helix harboring half a Me^2+^ BS in its center (omitted for clarity), shows synchronous downward progressions of h1b and h6a through the OO state, apparently mimicking the coordination of a bound substrate Me^2+^ ion. 

The relocations of h1b and h6a terminate simultaneously while h1aC begins moving toward h5, as demonstrated by the AF-A0A1P8Q6E3-F1 switch intermediate. This transition completes through the IO state with a further outward displacement of h1a, perhaps more realistically modeled using aCF while absent from the N-terminally truncated 5M94 structure. 

To validate MCb model collection as a set of possible conformers, select models were binned into candidate OO, IO and intermediate categories (Figure 2) to compute consensus networks of interacting amino acids (WebPSN). These analyses associated one community of networked residues with each orientation: OO (5M87 c7: h1a, h5, h8) and IO (5M94 nc18: h6a, h10, h11 (Appendix A)). The visible timing of creation/disruption of these networks in the series of models used indicates that, as expected for membrane carriers, occlusion of the external Me^2+^ binding site is coordinated with the opening of the inner gate. 

Another community of networked residues that evolves during progression toward and throughout the IO state (limon intermediate and orange-to-red ensemble) links h1b, h5, h7 and h8 (Appendix A). The limon intermediate maintains an OO connection between h1b, h7, h5 and h8, which is reduced to h1b, h7 and h5 once the MCb carrier has switched to the IO state. This community may help h5 tilt to open the inner gate as h1b appears connecting (via h10) outer gate closure (h6a, h10, h11) to opening the inner gate (h5, h7, h8).

Two additional communities of networked residues evolve either (i) during progression toward and throughout the IO state—part of the Slc11 H^+^ network (Appendix A)—implying it is dynamic as MCb conformation transition proceeds, or (ii) during forward switching between the OO and the IO state (Appendix A), which suggests a direct role for h6b. Based on these data, it was concluded that, in the absence of the Me^2+^ substrate [20], AF2/CF interpret MCb diversity and intrinsic dynamics to confidently predict a collection of carrier cycle intermediates.

### 2.2. Site-Directed Mutagenesis Demonstrates the Functional Significance of MCb Q5HQ64 CF Modeling

Given a suite of valid conformers populating intermediate steps during the MCb carrier cycle, it should be possible to induce in silico conformation exchange using appropriately targeted mutagenesis (i) to identify sites provoking MCb conformation exchange and (ii) to probe the impact of defined MCb communities of networked residues on this mutation-induced conformation switch.

The evolutionary concept of ‘Slc11 synapomorphy’ (Figure 6) entails a novel character shared by all the members of this family. This set of synapomorphic amino acid residues, products of type ii co-evolutionary rate shifts, was previously proposed to contribute to co-substrate binding and carrier gating [12]. Several members of Slc11 synapomorphy were found in communities of networked residues that evolve during OO-to-IO MCb carrier transition: two h6 sites involved in the MCb conformation switch (cf Appendix A) and both h10 and h11 sites mobilized for outer gate closure (cf Appendix A). Also, relaxing Slc11 synapomorphy definition would include h1a V47/-site which was linked to inner gate opening (Appendix A).

These data support that the 3D network of Slc11 synapomorphy may control carrier conformation transition. To examine this possibility, this set of synapomorphic residues, including the Me^2+^ BS, was targeted using select combinations of mutations that mimicked ancestral type ii evolutionary rate shifts.

To start, a model IO conformer (Q5HQ64) was chosen aiming to mimic the backward switch toward the OO state. Q5HQ64 was taken since its AF2 model (magenta, Figure 2) was closest to 5M94. The CF pdb top model of Q5HQ64, based on the local confidence score (predicted local distance difference test, pLDDT), was obtained using AF2 trained model2 (Appendix A). Although not ideal, because it shows minimal opening of the inner gate, CF pdb model2 displayed per-residue RMSD from reference MCb structures 5M87 and 5M94 indicating that it remained part of the IO orange-to-red ensemble (Figure 2, closer to orange CF pdb model than magenta AF2 model, data #1, Appendix A) and thus suitable to model a mutation-induced conformation backward switch from the IO state to the OO state.

Q5HQ64 h3 mutation A131Y G135N (h3 YN) is key to switching conformation backward to the OO state since it cooperates with additional mutations in either h1, h6a, h6b, h7 or h11 to trigger broad scale deviation of the per-residue Ca RMSD from the native Q5HQ64 IO conformation (Appendix A). The profiles of these mutation-induced OO states are conserved among various mutants, encompassing h1b-h2N, h5-l5/6-h6a, l7/8-h8N, h9C-h10N and h11C. Coordinated targeting of Q5HQ64 h3 (YN) and additional Slc11 synapomorphic sites therefore mobilized the same helical segments that were noted during the transition of native MCbs from the OO state to the IO state.

The influence of h1 Me^2+^ BS mutations depends on the conformation induced by the genetic background on which they were tested: absent in wt (data #2, Appendix A) and in the conformation-silent h6 VY mutation (Appendix A) but large and positive in the conformation switch conducive background (h3 YN); small and negative in a switch intermediate (VY YN, i.e., h3 YN h6 VY, Appendix A); and virtually absent in more fully switched OO conformation (h3 YN h6 VY A228T, VNT). These data suggest epistatic interactions consistent with a direct role of h1 Me^2+^ BS in Q5HQ64 conformation exchange.

Combining both h1 Me^2+^ BS mutations reversed some of the effects induced by each mutation assayed separately (Appendix A): it abrogated their strong positive interactions with YN, annulated their weak negative interaction with VY-YN and impacted negatively h7 N277T augmented VY-YN background without affecting the VY YN A228T-induced conformation switch. This indicates that h1 Me^2+^ BS mutations interact with one another, and this interaction depends on Q5HQ64 conformation. In addition, Q5HQ64 conformation switched backward to the OO state (VY YN A228T, VNT) fits between native MCb conformers green and cyan (Appendix A, inset, and Figure 3). These analyses support that Slc11 synapomorphy is instrumental to MCb carrier cycling and indicate an intrinsic role of Slc11 Me^2+^ BS in MCb conformation transition.

### 2.3. Central Role of h3 in MCb Q5HQ64 In Silico IO-to-OO Conformation Exchange

Q5HQ64 h3 YN mutation also cooperates with h10 mutation combination S393A L397N S398G (ANG) albeit moderately (Appendix A), while in the VY YN background, ANG produced a rearrangement slightly deviating from the profile of another triple mutant h6 A228T, h7 N277T and h11 N442G (Ttg), in the area of inter-helix contacts between h3 and h10 [6,7]. Also, h10 ANG cooperated with Ttg in the YN background, but this was not obvious in the VY YN background. In addition, h10 ANG negative influence in the VY YN background was not corrected by either h6 A228T, h7 N277T, h11 N442G (Appendix A) or their paired combinations (Appendix A), which suggests some role for h3 h10 inter-helix contacts.

Moreover, in both YN and VY YN backgrounds, combining h10 ANG with h11 N442G in the absence of h6 A228T reduced the per-residue RMSD of the resulting model (Appendix A), implying a limited conformation switch compared with h10 ANG only. Both h10 S398 and h11 N442 are constituents of MCb 5M94 IO community c11 [nc18] (Appendix A, left), and their combined mutation may impair outer gate function. Together, these modeling data therefore suggest a direct interaction between h3 and h10 that connects the MCb outer gate to the Q5HQ64 conformation switch. 

The YN mutation places Q5HQ64 h3 at the nexus of inter-helix connections involving other Slc11-specific sites whose combined mutation switches conformation to the OO state back from the IO state. Helix 3 limited motion during forward OO-to-IO transition (e.g., Figure 3) suggests that Q5HQ64 h3 mutations A131Y G135N (YN) may interfere with more mobile elements, such as the Me^2+^ BS (h1 D54, N57 and h6a A228) and/or h10 sites, to favor conformation exchange. The introduction of the orphan outgroup substituents YN deviates h3 locally, possibly relieving interactions normally formed in the IO state with h1, h6 and h10 and therefore easing the model switch backward to the OO state. 

The role of h3 in the Q5HQ64 conformation switch was questioned by (i) testing an alternative alignment of the h3 segment and consequently targeting different sites, whose mutagenesis remained conformation-silent (Appendix A), and (ii) considering the Slc11 outgroup perspective on h3 original alignment to mutate additional sites (Figure 7). Combined, six h3 mutations (GYGNGG) sufficed to induce maximal broad-scale deviation (independent of h6 mutation VY). Compound mutants VY YN 228T and GYGNGG exhibit similar OO profiles that differ at the h3 and h10 contact area (Appendix A), suggesting intrinsic flexibility allowing local compensatory rearrangement between these two helices. The data imply that altering h3 topology or curvature can switch Q5HQ64 predicted conformation from IO to OO.

### 2.4. MCb 3D Models Are Flexible and Share Alternate Communities of Networked Residues

A second round of mutagenesis aimed at testing the role of the communities of networked residues, previously identified using ensembles of native MCb models in different states (OO, intermediate, IO), in the in silico conformation switch modeled between wt Q5HQ64 (IO) and the VNT mutant (VY YN 228T, OO).

Site-directed mutagenesis intended to maintain either the inner gate open, using bulkier side chains, or the outer gate closed, providing a possibility for a salt bridge, prevented full-scale conversion of the IO state into the OO state and proper rearrangement of the Me^2+^ BS (Figure 8). The data indicate the functional significance of the targeted communities of networked residues both in Q5HQ64 conformation exchange induced in silico and more generally, in model MCb carrier cycling.

In contrast, mutagenesis interfering with sites of the Slc11 H^+^ network had little impact (Appendix A), suggesting that either the substitutions assayed were not effective or the sites tested did not contribute to intramolecular rearrangements leading to conformational change. In addition, targeting a community of hydrophobic residues (h1, h5, h7, h8) possibly involved in h5 tilt toward the IO state (Appendix A) showed that few among the substitution combinations assayed limited the amplitude of per-residue RMSD in the areas of h1b and l5/6 and h6a (Appendix A). These combinations represent the consensus sequence of either outgroup (orphan or NRMT) or aN (aN-I or aN-II) clades, which constitute the root and tip of the Slc11 family tree, respectively (Figure 1), hence suggesting possible significance. 

To verify that properties of the Q5HQ64 CF pdb model apply to distinct MCbs, another sequence (WP_002459413, cf Figure 2, 83% identical to Q5HQ64) was shown to respond similarly to the VNT mutation (Appendix A). In addition, A0A380H8T1 (the closest Uniprot relative of *S. capitis* 5M94, 90% id with Q5HQ64), which produced an IO CF model like the Q5HQ64 model, was subjected to identical mutations (Appendix A). Some results deviated from the findings of the Q5HQ64 study, such that h1 Me^2+^ BS mutations did not cooperate with h3 YN for broad-scale rearrangement (Appendix A) and VY YN strong sensitivity to h1a D48G mutation (Appendix A). Yet, A0A380H8T1 h3 YN mutation required cooperation with h6 VY mutation to switch wt IO conformation back to the OO state, and the resulting per-residue RMSD was also globally enhanced by combining with h6a A223T. Hence, native IO conformers from distinct MCb respond similarly to compound mutagenesis targeting sites part of Slc11 synapomorphy by switching back to the OO state.

Modeling of both A0A380H8T1 and WP_002459413 was also highly sensitive to targeting the MCb OO community (5M87 c7: h1a V50, h5 V177, h8 N334), which strongly antagonized the VNT (VY-YN-223/4T)-induced conformation switch (Appendix A), having direct effects on h1, h5 and h8 topology (profile resembling that of A0A380H8T1 YN D48G, Appendix A). Furthermore, mutating sites of the MCb IO community (5M94 c11 [nc18]: h6a L223, h10 S398, h11 N442) impacted VNT-induced conformation, as previously observed for Q5HQ64, by decreasing the overall amplitude of the switch (Appendix A). Hence, mutually exclusive communities of networked residues identified among separate pools of native MCb conformers, either OO or IO, control the in silico conformation switch from the IO state to the OO state of distinct MCb templates.

Comparing native IO and mutation-induced OO conformers obtained for distinct MCbs (Figure 9) shows pairs of superimposable structures distinguished by coordinate movements of sets of helix segments: (i) h1b-h2N [incl. l1/2] and h5-[l5/6]-h6a and (ii) h10N-h9C [incl. l9/10]. Synchronized motion of these segments may complete occlusion of the external Me^2+^ BS while locking the outer gate. The MCb IO-to-OO conformation switch also shows a small displacement of h1aC while h6b remains still, as seen in MCb progression from the OO to the IO state (i.e., between green and orange intermediates, Figure 5). This suggests that h1aC’s early move may initiate the opening of the inner gate (unlocking). The ‘hash module’ helices h3 and h8 in contrast show minimal deviations (except for areas of h3 A-to-Y and G-to-N (YN) mutations and the h8N end), implying that they may contribute to the carrier conformation switch by interacting with residues from more mobile parts of the structure (e.g., h1 and h10 for h3 and h1a and h5 for h8).

AF2/CF modeling thus exploits MCb diversity and intrinsic dynamics to confidently predict conformers likely representing discrete steps in carrier cycling. Slc11 synapomorphy comprises MCb conformation driving sites, including notably prominent contributions of h3, h1/h6 Me^2+^ BS and h10 residues. Specifically, Slc11 synapomorphy is coordinately rearranged during the MCb carrier switch (Figure 10A); subsequently, Slc11 synapomorphy deviates minimally albeit significantly (e.g., area of D54, Figure 10B).

Notably, native MCb CF pdb models (Q5HQ64, A0A380H8T1, A0A853V1J2, >80% id) were superimposable (e.g., Figure 10C) and highly similar to AF-A0A853V1J2 (D54 area, Figure 10B). The three MCb VNT CF pdb models are also highly similar, resembling native OO carriers (e.g., AF-Q74JG3, CF pdb-WP_103371048, Figure 10A) as they mimic conformation changes taking place during the carrier switch (A, inset) despite local deviation of h3 and h1a (C).

While CF pdb predictions of related MCbs (>80% id) model a single conformer at an early step following the OO-to-IO forward switch (e.g., WP_002459413 Figure 2), their respective AF2 models depict either similar or ensuing steps to open the inner gate, in a sequence-specific manner (AF-A0A853V1J2 < A0A380H8T1 < Q5HQ64), with coordinated local deviations of h1a, h5, h7 and h8 (Appendix A) that correspond to two MCb residue communities previously involved with h5 motion (Appendix A, left panel, and Appendix A). Conversely, the MCb residue community implicated in h10 motion (Appendix A) evolves minimally in post-switch steps, suggesting that locking the outer gate is concomitant with complete occlusion of the external Me^2+^ BS (Appendix A).

These data suggest that MCb OO-to-IO forward transition is modeled in two processes: (i) a substrate-selective conformation switch that may unlock the inner gate and (ii) post-switch substrate-dependent intracellular release (completing both Me^2+^ BS occlusion and outer gate locking while opening the inner gate). Detailing the motion of MCb residue communities controlling h5 movement supports this interpretation (Appendix A). Substantial rearrangement during the carrier switch could unlock the inner gate (Appendix A) while in subsequent steps, h1b and h7 contribution to shift h5 appears selectively reduced (perhaps forming a pivot point) while that of h1a is sustained (Appendix A). 

In sum, Slc11 synapomorphy plays a key role in articulating the MCb carrier substrate-selective conformation switch and contributes to substrate-dependent intracellular release. As such, it constitutes a set of useful target sites to appreciate structural and functional differences between Slc11 phylogroups in general and MCb and MCg sister groups in particular.

### 2.5. MCg1 Carrier A0A149PND7 Undergoes Similar In Silico Conformation Switch

Although MCb and MCg genes derive from a common ancestor [12], none of the MCg top models produced by AF2 or obtained through CF pdb modeling grouped with either an IO or OO MCb conformer (5M94 and 5M87, respectively; Figure 11). CF nt modeling using AF2 training model2 gave a candidate IO conformer for A0A149PND7 (MCg1). Using aCF (advanced settings previously used for MCb Q5HQ64) yielded a similar IO conformer with a slight improvement in the inner gate opening, which was later used to characterize additional MCg1s.

CF nt modeling with default parameters was used to study the impact of site-directed mutations on MCg1 A0A149PND7 predicted conformation because this setting appeared more suitable than advanced CF parameters in prior studies with MCbs.

To examine the role of h3 in the MCg1 carrier conformation switch, identical mutation combinations were tested in parallel using MCg1 A0A149PND7 and MCb Q5HQ64 templates (Appendix A). Unexpectedly, A0A149PND7 h3 A119Y G123N (YN) mutant adopted a fully switched OO conformation, marginally affected by additional h3 mutations, which was further stimulated combined with h6 A216T M219V H221Y mutation (VNT216). The data indicate that similarly to MCb Q5HQ64, A0A149PND7 h3 is key to switching carrier conformation albeit mediating its role in a distinct way. Testing h3 YN mutation with h10 ANG ^+^/_−_ h11 N422G or D (Appendix A) confirmed this interpretation: cooperation between Q5HQ64 MCb h3 YN and h10 ANG (Appendix A) was matched by A0A149PND7 h3 YN tolerance for h10 ANG, and in both instances (cf Appendix A), h11 N422G regulated the carrier conformation switch.

To evaluate the contribution of h1 Me^2+^ BS in the MCg1 A0A149PND7 conformation switch, the mutations D42G and N45T were assayed alone and combined in distinct backgrounds YN, VY and VY YN (Appendix A). The negative impact of D42G combined with h3 YN and of both D42G and N45T combined with h3 YN h6 VY (VN) was noted. These results support that interaction between h3 YN and h6 VY mutations alters MCg1 A0A149PND7 modeling sensitivity to h1 mutations, as previously observed with MCb Q5HQ64 (cf Appendix A). The data imply that Slc11 synapomorphic residues maintained a prominent contribution to the carrier switch mechanism despite the evolutionary divergence of MCb and MCg1.

Targeting the mutually exclusive communities of networked residues identified in ensembles of MCb homologs (Appendix A), which controlled the MCb VNT-driven conformation switch (Figure 8 and Appendix A), had a mainly local impact and little effect on the overall conformation of MCg1 A0A149PND7 VNT (VY YN 226T, Appendix A), e.g., deviation of h5 and h4 (A, D), displacement of h6a, h10 and h11 (B, E) or slight outward deviation (tilt) of h5 and minimal impact on h1b, h6a and h10 (C, F). These results suggest differences in carrier intramolecular dynamics consequent to divergent evolution between the sister groups MCb and MCg1.

To explore that aspect further, A0A149PND7 secondary suppressor sites of the YN-driven conformation switch were sought by examining the YN mutant model. Possible sites of direct contact were located on h1, P43, and h10, L370 and Q374. Mutating P43A alone or combined with N45G had essentially no impact on modeling (Figure 12A,B). To obtain a substantial effect, it was necessary to mimic the NRMT sequence in this area, with the four-residue exchange I41N P43A N45G W46I (NAGI).

Turning to h10, reducing Q374 side-chain volume reversed the YN-induced conformation switch, at least partly, by better accommodating h3 A119Y substitution (Figure 12C,D). Combining both h1 and h10 mutations in the A0A149PND7 YN background had additive effects resulting in close to wt IO conformation (Figure 12). The data indicate that MCg1 A0A149PND7 h3 YN mutation induces a conformation switch by altering inter-helix contacts between h3, h1 and h10.

Indeed, YN compound mutant 394G, but not NAGI, was rescued by h6 VY mutation to produce YN-like broad-scale RMSD (Appendix A). The YN 394G reduced conformation switch was also reverted to YN levels when combined with h6 A216T, h11 N422G or both but not h7 N264T. And h7 N264T disrupted cooperation between A0A149PND7 mutations YN (h3), A216T (h6), 394G (h10) and N422G (h11) (Appendix A). Lastly, substituting h11 N422 regulated cooperation between h3 YN, h10 Q394G and h10 ANG (Appendix A).

Introducing h10 Q394G mutation in the MCg1 A0A149PND7 YN mutant therefore restored cooperative behavior between mutations targeting synapomorphic sites (h3 YN and h6 VY, h6 A216T, h11 N422G). The similarity of the resulting conformation changes with those previously observed with Q5HQ64 indicates that the MCb switch mechanism was conserved in A0A149PND7 MCg1. Yet, some data obtained studying the A0A149PND7 conformation switch, such as the negative influence of h7 N264T mutation and distinct behavior of h3 mutants, suggest also a degree of functional divergence.

### 2.6. Conserved Switch Mechanism in Extensively Divergent MCg1 from Bordetella-Achromobacter

To determine whether A0A149PND7 divergent response to targeted mutations is shared among MCgs, another wt model in IO conformation was necessary. CF nt models obtained for MCgs were re-examined in Dali AAA comparisons, including as reference models those from wt MCg1 A0A149PND7 (IO) and its converted form, A0A149PND7 VY YN 226T (OO), together with the MCb pair of structures OO (5M87) and IO (5M94) as the outgroup instead of the pair of MA structures IO (6D9W) and OO (6D91) previously used. 

These reanalyses pointed to another MCg1 sequence (s011, D4XFA5), which yielded both IO and OO conformers depending on the training model used (CF nt rr3-5 and CF nt rr1-2, respectively, Figure 13A,B). Notably, D4XFA5 VNT-induced OO conformer showed minimal sensitivity to dual mutation combinations targeting both residue networks involved in MCb carrier gating (i.e., alternate OO community h1a V50, h5 V177 and h8 N334 and IO community h6a L223, h10 S398 and h11 N442). In contrast, each mutation combination inhibited slightly a native D4XFA5 OO profile and showed a significant impact when assayed together. These data suggest that alternate communities of networked residues involved in MCb gating may play similar roles in MCg1 carriers.

To verify h3 central role in the MCg1 conformation switch, the mutation series assayed previously with MCg1 A0A149PND7 and MCb Q5HQ64 (cf Appendix A) was tested using MCg1 D4XFA5. These data (Figure 13D) show variation on a common theme, wherein the triple mutation GGG triggers defective broad-scale rearrangement that excludes substrate-selective elements, which become mobilized upon combination with YN mutation. The results imply dual roles of D4XFA5 h3 in carrier cycling by providing a conformation switch instructive signal and linking it to the reorganization of the Me^2+^ BS.

Modeling D4XFA5 MCg1 mutants resembled more that of MCbs than MCg1 A0A149PND7, showing the cooperation of h3 YN and h10 ANG mutations to produce a broad-scale conformation switch modulated by h11 D421 mutation (Figure 13E, cf Appendix A). Functional similarity of D4XFA5 MCg1 and Q5HQ64 MCb also includes cooperation between h3 YN and h6 A216T or h11 D421G and between h3 YN and h6 VY (VN, Appendix A, cf Appendix A), though the VN profile was insensitive to several mutations (Appendix A). And similarly to A0A380H8T1 MCb (Appendix A), D4XFA5 modeling showed a lack of interaction between h3 YN and the Me^2+^ BS mutations tested (Appendix A).

As with both MCbs and A0A149PND7 MCg1, the impact of Me^2+^ BS mutations depends on D4XFA5 conformation resulting from the genetic background used: both D42G and D42G N45T mutations abrogate the VY YN-induced conformation switch (Appendix A), but D42G has no effect in the VNT background (Appendix A), implying functional differences between VN- and VNT-induced conformers. Along similar lines, mutation combinations inducing similar switches (YN A216T and YN D421G, Appendix A) are differently impacted by adding h1 D42G (Appendix A).

CF nt modeling of distinct MCg1 carriers therefore recapitulates the conformational changes established previously for MCb carriers, with similar IO-to-OO conformation switches coordinately mobilizing the same structural elements and induced by similar combinations of synapomorphic mutations (Appendix A). Divergence among the MCg1 carriers studied (41.4% id) influences the positions of h1a and h6b, as well as h3 and h8, which both seem species-specific rather than conformation-dependent. Yet, h1b and h6a movements appear consistent with those previously observed for MCb templates as well as the motion of the h9C-h10N ensemble. A limited shift of h1aC below the Me^2+^ BS is also discernable, together with h5 tilt. The data imply that similar principles govern both MCb and MCg1 carrier cycling.

It seems also possible that MCg1 divergence altered carrier function. Examination of multiply aligned sequences and superimposed models from MCb and MCg1 sister clades reveals changes in the MCg1 H^+^ network that modify the topology of h4 (Figure 14). MCg1 h4s lack a conserved D residue part of the (pre-) eukaryotic Slc11 H^+^ network (shared by MntH H and Slc11 eukaryotypes: aNs, pNs and MCs (Figure 1)) [12]. CF modeling shows that the absence of this D moiety is compensated by arranging in its 3D position the adjacent T side chain and consequently deviating the position of h4 (Figure 14, inset). This local rearrangement is also observed with native MCg1 models in the IO state (data #3, Appendix A). Such intramolecular reorganization may preserve the transmembrane H^+^-network geometry by altering the MCg1 carrier shape and perhaps its activity.

In this context, further evolution in the MCg1 clade may bear functional significance. Given that D4XFA5 MCg1 represents one of the most divergent sequences included in the Slc11 family set [12], it was surprising that A0A149PND7 displayed the most distinctive modeling properties. To investigate this further, MCg1 phylogeny was detailed. 

The extent of divergence of the D4XFA5 MCg1 cluster is demonstrated by maximum likelihood phylogenetic analysis of NCBI microbial sequences collected using MCg1-selective PHI-Blast search (Appendix A). The tree presented suggests MCg1 could emerge in Betaproteobacteria (BPB) of the Burkholderiaceae family (310 organisms) (*Burkholderia* > *Paraburkholderia* > *Caballeronia* > *Pandoraea* > *Cupriavidus*) from a common ancestor that was shared with MCg2 (e.g., A0A149PND7). Notably, the D4XFA5 MCg1 divergent cluster shows non-overlapping taxonomic distribution, restricted to BPB spp. of the Alcaligenaceae family (90 organisms, *Achromobacter* > *Bordetella*) plus another BPB (Rhodocyclaceae), a few GPB (Xanthomonadales) and one APB.

Zooming in on the D4XFA5 MCg1 cluster reveals that the *Bordetella* genus could harbor this gene since its origin (Appendix A), as MCg1 phylogeny supports previously established species relationships in this genus, positioning *B. ansorpii* basal to the divergence of the sister clusters *B. flabilis*-*B. bronchialis* vs. *B. avium*-*B. hinzii* [21]. In addition, the *B. flabilis*-*B. bronchialis* cluster includes two *Achromobacter* seqs, while the remaining forms a sister cluster. These data point to a likely origin for the *A. piechaudii* D4XFA5 encoding gene because homologs from *A. aestuarii*, *A. agilis* and *A. veterisilvae* stand in basal positions, confirming prior studies based on conserved genes [22,23]. Phylogenetic reconstruction of the D4XFA5 MCg1 cluster also apparently discriminates clinical from environmental specimens of *Achromobacter* [24,25,26], implying clock-like behavior that indicates that D4XFA5 MCg1 is a functional protein.

Assuming that h4 and internal H^+^-network reorganization in MCg1 ancestor introduced steric constraints, as evidenced with A0A149PND7 models, their relative absence deduced from D4XFA5 modeling suggests divergent epistasis in this cluster. Arguably, the extensive divergence of the D4XFA5 cluster served to restore more canonical carrier dynamics.

Regarding residues that are identical among MCg2 and D4XFA5 MCg1 clusters but distinct in the A0A149PND7 cluster, the results pointed out six sites spatially connected in two sets of residues from either h1b, l1/2 and h7 or h4 and h8 (Figure 15A,C). Considering group-specific residues, 45 sites scattered across transmembrane helices showed a high level of spatial colocalization (Figure 15D,F), linking as well h1a and h7 and strengthening h4 and h8 connection together with h5; prominent interactions among h3 and h9 and l9/10 are very clear, as well as several instances of colocalizing sites involving distinct helices.

Molecular evidence thus supports the view that the A0A149PND7 MCg1 cluster may represent an evolutionary intermediate that could be co-opted and further adapted in the emerging *Bordetella-Achromobacter* genus, for instance. Together with modeling and mutagenesis data showing distinct structural dynamics for A0A149PND7 compared to D4XFA5 and MCbs, it seems plausible that the divergence of the D4XFA5 MCg1 cluster in *Bordetella* aimed at tuning carrier structural dynamics, including the MCg1 neo H^+^ network.

### 2.7. Slc11 Synapomorphy: A Conserved Framework Articulating Carrier Cycling in Divergent 3D Contexts

Though modeling of VNT (VY YN A228/223/216T) mutants displayed a similar backward switch toward the OO state for both MCbs and MCg1s (Figure 14), template sensitivity to the mutations tested varied. Notably, steric interference limited A0A149PND7 MCg1 conformational plasticity, and both MCg1 templates exhibited limited or negative interaction between h3 YN and h7 264T and displayed altered h4 topology. Global variations in carrier shape were thus examined by correspondence analysis of Dali all-against-all 3D structural comparisons, using pairs of alternate model states (OO and IO) representing the sister groups MCb and MCg1, plus the reference PDB structures from MCb and the distant phylogenetic group MntH A (Figure 16).

The results show parallel meta-ensembles (combining both IO and OO states) that clearly delineate MCb models and structures apart from MCg1 models and MA structures (Figure 16A), a result seemingly at odds with Slc11 phylogeny (Figure 1). IO and OO states segregate in both meta-ensembles, and mutagenesis-induced in silico switches from IO to OO conformation appear similar for MCb and MCg1 models. These data support common principles governing MCg1 and MCb carrier cycle while divergent 3D features distinguish MCb from MCg1 structures, e.g., MCb sequence divergence and structural variation in l7/8 (data #4, Appendix A) and MCg1 rearranged h4 and H^+^ network. In other words, the data suggest conservation of the carrier mechanism of conformation exchange within divergent 3D contexts.

Focusing on the structural divergence between MCb and MCg1, per-residue RMSDs were calculated between VNT OO mutants, using Q5HQ64 VN228T as reference (Figure 16B). The profile depicting structural variation among MCbs (gray) identifies peaks in the areas of l7/8, l5/6, l1/2, l9/10 and h11C, which testify of MCb flexibility. In contrast, MCg1 profiles (in red) appear drastically different, with only a few areas showing limited deviations and covering the central part of h1 and h6 (Me^2+^ BS), h3, h5, h7, h10 and h11. Hence, MCb and MCg1 VNT-induced OO models display extensive structural differences supporting their grouping in distinct meta-ensembles (Figure 16A).

Yet, areas where MCb and MCg1 models deviate least map close to the sites forming Slc11 synapomorphy (respectively, Q5HQ64 h1 D54, N57 and h6 A228, M231, H233; h3 A131, G135; h7 N277; h10 S393, L397, S398; h11 N442), as well as the h5 putative pivot point (h5 F188, cf Appendix A). These same areas display similarly minimal deviation when native models are compared (data #5, Appendix A). Accordingly, the 3D arrangement of Slc11 synapomorphic sites remained constrained as MCb and MCg diverged and could therefore preserve Slc11 mechanism of conformation exchange within evolving 3D contexts. Indeed, the 3D superposition of MCb and MCg1 OO models (VNT mutants) shows little deviation in the helix backbone vicinal to Slc11 synapomorphic sites (Figure 14), implying that conserved 3D arrangement of these specific areas is instrumental to both MCg and MCb carrier cycling.

Other areas showing 3D morphism between MCb and MCg1, such as h3N and l6/7 (Figure 16B), may contribute as well to maintaining the 3D integrity of the Slc11 functional network. MCb sequence divergence may promote phylogroup-specific structural variations (e.g., l2/3, h4-l4/5) or even more specific structural determinants (e.g., extensively divergent l7/8 and local deviation of l9/10) that together contribute to the heterogeneity of carrier shapes. On the other hand, 3D neighboring of MCg1 models and solved MA structures (Figure 16A) could reflect structural convergence resulting from the MCg1 path of evolutionary divergence (Appendix A).

Based on the data produced through this study, a simple interpretation is proposed: Slc11 synapomorphy forms a conserved 3D framework that articulates conformational transition during carrier cycling and whose geometry may be adapted to distinct contexts, because of specific divergence. Indeed, the fixation of multiple evolutionary rate shifts has the potential to create high-order epistasis [27]. Slc11-specific residues would act as functional nodes of a 3D network whose edge dynamics may vary between phylogroups if the respective positions of the nodes were coordinately maintained and/or adapted. Under these assumptions, MCb could evolve into highly flexible carriers, providing plastic loops, while in comparison, MCg1 adapted to steric constraints (Figure 10) and/or helix distortions (Figure 14).

Together, the results obtained argue in favor of functional divergence between the sister groups MCb and MCg because the Slc11 core functional network persisted embedded in distinct 3D contexts, which in turn altered intramolecular dynamic properties through functional epitasis depending on local variations in steric hindrance and helix topology. MCg1 D4XFA5 cluster epitomizes functional divergence that correlates with the emergence of the *Bordetella*-*Achromobacter* genus, implying potential gain of function.

## 3. Discussion

The biological significance of the present study can be summarized as follows. AF2/CF modeling demonstrates the structural divergence of sister groups of bacterial Slc11 carriers of eucaryotic origin, MCb and MCg, which were previously associated with horizontal dissemination amid distinct ecological niches. Coupled with phylogeny-informed site-directed mutagenesis, AF2/CF modeling further reveals the conservation of a 3D network articulating similar conformation changes within the divergent architectures of MCb and MCg1 carriers. MCg1 site-specific divergence shows steric hindrance altering carrier shape and dynamics and divergent epistasis that correlates with the emergence of novel bacterial genera. On the other hand, MCb loop plasticity fosters conformational flexibility. The data suggest different carrier dynamics implying divergent adaptation of H^+^-dependent Mn^2+^ uptake in distinct environments.

Considering the Slc11 protein family as a monophyletic clade and applying the apomorphy concept to type ii evolutionary rate shifts (i.e., site-specific replacements of a conserved residue by a novel, also conserved residue) led to short-list residues to be targeted to probe the carrier conformation switch and permitted direct comparisons across sister clades. Clade-specific paths of divergence were distinguished, wherein a 3D framework articulating the carrier conformation switch (Slc11 synapomorphy) is maintained at the cost of local structural deviations impacting carrier shape and dynamics. Combining AF2/CF modeling tools with clade-specific site-directed mutagenesis thus holds the potential to better understand sequence/structure/dynamics/function relationships in families of integral membrane proteins.

MCb plasticity and conformational flexibility likely aid AF2/CF in exploiting sequence diversity to model an array of structures representing plausible carrier conformers. Several results established these conformers as functional intermediates: (i) binning models of different origins into alternate (OO or IO) or in-between carrier states led to the identification of communities of networked residues that evolve between these three stages; (ii) some of these communities evidenced direct contribution to conformation exchange after site-directed mutagenesis combined with CF pdb modeling using distinct MCb templates; (iii) the mutagenesis strategy targeting Slc11 synapomorphy produced an in silico backward conformation switch mobilizing the same helical segments identified during the forward transition of native model conformers; (iv) modeling of site-selected MCb mutants showed dynamic responses revealing the central role of h3 linking outer gate closure (h6a, h10, h11) to conformation exchange (h1, h6, h7) and inner gate opening (h1a, h5, h8), including inter-dependent contributions of h1 Me^2+^ BS that hinge on the carrier conformational state. These results appeared in line with the basic concepts of carrier-mediated transport [19].

Distinct MCg1 models underwent a similar switch, involving the same helical segments as in MCb models, which signifies a conserved switch mechanism, including gating elements, despite divergent evolution. But MCg1 templates’ sensitivity to targeted mutagenesis differed, evidencing steric hindrance in one instance and apparently balanced by extensive epistatic divergence in another case. MCg1s share a point mutation in h4 that alters the Slc11 H^+^ network and appears compensated by h4 distortion and deviation. Molecular analyses indicate extensive evolutionarily coupled divergence that correlates with the emergence of the *Bordetella* genus.

An example of conserved functional connection among MCb and MCg1 models, yet subjected to MC group-specific structural constraints, is given by h1aC motion during the carrier switch. AF2/CF pdb models of MCbs show a strong phylogenetic component (Figure 17).

The limon intermediate conformers (e.g., Figure 2) form a divergent cluster with mutations in h8, among others, including the residue from the networked community (h1a, h5, h8) implicated in inner gate opening (h8 N to S, Figure 8E). Both h8 and h5 sites are rearranged in the limon intermediate conformers while most of the Slc11 synapomorphy display the expected architecture (except h3 and h7, Appendix A). The motion of h1aC, shown by MCb limon AF2 intermediates and between native IO and VNT mutation-induced OO CF models of both MCbs and MCg1s, therefore captures an event prior to inner gate opening (unlocking). MCb divergence may impair the modeling of h8 interaction with h1aC and result in the prediction of intermediate conformers.

Correspondence analysis of multiple pairwise 3D alignments confirms that MCg1s and MCbs undergo similar mutation-induced conformation exchange; yet, extensive structural variation affecting carrier shape underscores divergence between these sister groups. On the one hand, a few areas in MC carrier structures show minimal deviation between superposed models of MCb and MCg1 in similar states (IO or OO), and these areas map close to Slc11 synapomorphic sites. Thus, Slc11 synapomorphy represents the 3D preservation of a functional network that articulates metal-selective conformational changes. On the other hand, helical segments that deviate between MCg1 and MCb counterparts represent potential structural constraints that may affect carrier shape and influence the dynamic response to mutagenesis targeting Slc11 3D functional network.

In this regard, it appears significant that extensive epistatic divergence of one MCg1 cluster restores a dynamic response to mutagenesis similar to MCbs vs. the other MCg1 cluster. Molecular data suggest that the MCg1 H^+^-network amendment/deviation of h4 was a founding event for this subgroup which originated from a common ancestor shared with the MCg2 subgroup. In spite of the structural constraints incurred by altering the carrier shape, it seems likely, based on taxonomic distribution, that MCg1 isoform represented by A0A149PND7 fulfills a biological role in the soil bacteria of the plant rhizosphere. It may thus be surmised that divergence of the MCg1 isoform related to D4XFA5 enabled adapting this role in the emerging *Bordetella* genus.

The functional importance of the MCg1 H^+^-network amendment/deviation of h4 is underscored by a broader evolutionary perspective. During Slc11 evolution, a T moiety appeared in the h4 of bacterial MntH A, expanding their internal H^+^ network. Then, in the common ancestor of archaeal MntH H and Slc11 eukaryotypes (Figure 1), the carrier incorporated, among others, an adjacent D residue preceding this h4 T moiety, hereby extending possible interactions between the transmembrane H^+^ network and the proton motive force driving Me^2+^ import [12]. MC descendants of prototype Nramp possess this h4 D residue, except all MCg1s. The h4 DT pair was most strictly conserved in pN-II, MCg2, aN-II and MCaU. The loss of h4 D is only observed in MCg1 and in aN-I. However, in aN-I *A. thaliana* Nramp1, the substitution of D for S apparently preserves both the H^+^ network and h4 topology (Q9SAH8). These evolutionary patterns add credence to the regulatory role of the h4 DT pair and support the functional significance of the MCg1-specific rearrangement of h4. This suggests that MCg1 structure preserves Slc11 H^+^ network at the expense of the carrier shape and perhaps by altering carrier activity.

By analogy, upholding of Slc11 3D functional core network (Slc11 synapomorphy) in fast-evolving clades, such as MCb and MCg1, may impact carrier shape and/or dynamics of transport. Assuming the latter is a driving force for diversification, distinct adaptive paths may lead to clades of carriers whose flexibility and dynamics differ widely. AF2 models for other MC groups (MCa and MCaU) resembled those of MCgs, apparently lacking the conformational flexibility displayed by MCb models [12]. Future studies will be required to determine whether this means additional variety in terms of MC carrier intramolecular dynamics.

From an evolutionary perspective, it is most parsimonious that MCbs acquired higher flexibility on their own, in response to selective environmental constraints. Examples of benefit and/or competitive advantage conferred by genes encoding MCbs have been reported [28,29], and carrier flexibility leading to conformational diversity may contribute to its evolutionary success [20,30]. Besides variations in loop length and sequence, interactions with water molecules may regulate carrier flexibility by forming a network acting to stabilize key residues and/or interaction with the Me^2+^ substrate [31,32]. Also, changes in protein dynamics upon metal binding could have important roles for substrate coordination by the membrane carrier and effective transport [33].

Slc11 prime substrate Mn^2+^ has shown multifaceted roles in bacterial adaption to stress, including during infection, and in metallo-cross talks that regulate metabolic homeostasis [15,17,34]. The MntH C sister clades studied here, MCb and MCg, derive from a common ancestor but display niche-selective, non-overlapping taxonomic distributions [12] that relate to several public health issues.

MCb predominates in Mn-centric Lactobacillales, including several pathogens such as *Enterococcus* and *Streptococcus* spp.; it is also prevalent in related bacteria utilizing both Fe and Mn (Bacillales), including *Staphylococci*. *mntH Cb* genes from lactic bacteria used in the dairy industry support metabolic activity [29] and competitive exclusion [28]. Participation of *mntH Cb* genes in adaptive responses coping with host nutritional immunity and acid and/or oxidative stress contributes to the virulence of the caries pathogen *S. mutans* [35], group B *Streptococci* [24,36] and the opportunistic pathogens *E. faecalis* [37] and *S. aureus* [38]. To what extent MCb roles owe to intrinsic structural diversity and flexibility awaits further investigation.

MCg is mostly found in iron-centric Proteobacteria from distinct classes (GPB, Pseudomonadales, and BPB, Burkholderiales) sharing many properties as environmental soil bacteria frequently found both in the plant rhizosphere and the human flora, including various spp. causing opportunistic infections in immunologically compromised individuals [39,40]. That divergence of the MCg1 D4XFA5 clade correlates with the emergence of both *Bordetella* and *Achromobacter* genera (Family, Alcaligenaceae; Order, Burkholderiales) is intriguing [41] and warrants further investigation because they include emerging pathogens associated with resilient nosocomial infections of the lung [25,42,43]. A causal link between MCg1-specific, environmentally constrained adaptation of H^+^-dependent Mn^2+^ uptake and bacterial pathogenicity, e.g., opportunist lung infection of persons with cystic fibrosis, remains to be addressed.

Deciphering the functional evolution of MC clades b and g thus bears the potential to gain important knowledge about Mn availability at distinct interfaces of host cell and microbe interactions. Bacterial species harboring several *mntH Cb* or *mntH Cg* genes are common and may favor diversification through recombination, yielding reservoirs of structural variants. Assuming that the h4 loss of D mutation altered MCg1 function, it seems plausible that it was compensated by an array of stabilizing mutations at various sites in the protein, fostering MCg1-specific divergence from its ancestral state [44]. Though epistatic drift through intramolecular interactions may render protein evolution less predictable [45], stressful and/or adverse environments exerting strong selective pressure can channel such a process [46]. This seems to be the case with the expansive divergence of the (D4XFA5) *Bordetella-Achromobacter* MCg1 clade wherein structural divergence suggests shifted constraints that could stabilize MCg1 functional evolution in these emerging bacterial spp.

The present work demonstrated multiple interactions revealed by mutating several sites that form Slc11 synapomorphy, suggesting that collectively, they may produce high-order epistasis and frame the evolution of this protein family [27]. Adverse growth conditions, e.g., iron deficiency, severe stress (heat shock, hydrostatic pressure, osmotic stress) that deplete cellular energy (both Mg^2+^ and ATP) or bacteriophage infection [12], may apply sufficient selective pressure to entrench H^+^-dependent Mn^2+^ import function within the Slc11 synapomorphy-based 3D framework. In turn, high-order epistasis could accommodate continuous adaptation to function-altering mutations in response to environmental changes, including the stepwise elaboration of Slc11 proton network, the loss of h4 D residue in MCg1 proteins and further evolution in the *Bordetella*-*Achromobacter* genus.

The framing of carrier evolution by high-order epistasis may represent a common property of the LeuT fold: the sheer number of different families it produced showing distinct substrate specificities and using an array of chemiosmotic mechanisms to energize transport, yet still sharing properties relating to carrier conformation transition, such as the inner gating of h5 and bending of h4, outer gating of h10 and bending of h9 or coordinated motion of h6a and h1b segments [5], together imply that the LeuT fold is highly evolvable, meaning intrinsically stable and robust to mutational epistasis [47]. As evolutionary pressure determines long-distance evolutionary coupling [48], distinct substrate specificities and/or chemiosmotic mechanisms could shape family-specific high-order epistatic pathways that sustained divergence in the APC superfamily. Maintaining a favorable energy landscape to be traversed during carrier cycling may constrain high-order epistasis [5], and a few residues articulating the carrier switch could foster fold evolvability and functional diversity.

To conclude, the divergent evolution of MCb and MCg1 affecting carrier shapes and dynamics may be viewed as alternative evolutionary trajectories resulting from different influences [49]. These include (i) historical chance for the initial founding of each clade and occurrence of function-altering mutations and (ii) high-order epistasis imparted by the 3D conservation of Slc11 synapomorphy, which articulates a substrate-selective carrier conformation switch. Prevailing within evolving 3D contexts, Slc11 synapomorphy is thus a reliable phylogenetic marker for function.

## 4. Materials and Methods

AF2 3D models for Uniprot MCb sequences were mined for candidate conformers like MCb reference structures 5M87 (outward open, OO) and 5M94 (inward open, IO) by screening pools of models using EBI all-against-all Dali (AAA) [50], plus a pair of *D. radiodurans* MntH A structures 6D91 (OO) and 6D9W (IO) used as outgroup [51]. Additional models generated through Colabfold (CF) modeling (v1.4 and v1.5.2) [52] using default parameters, including pdb template (CF pdb) or without template (CF nt), and Advanced CF modeling (aCF, parameters: MSA meth Jack hammer, pairwise MSA option 75% coverage, 15% id, max_msa_clusters = 32, max_extra_msa = 64, no use of ptm sampling, num_ensembl 1, max_recycle 1, use is-training, num_sample 1, refine/Amber) [53] were used to explore candidate MC IO models.

A temporal sequence of possible structural changes occurring during OO-to-IO transition was inferred from graphical representations of per-residue Ca root mean square deviation (RMSD) relative to each MCb reference structure (5M87 and 5M94) for each of the conformer selected, established using 2StrucCompare [54]. Structural changes were visualized using Pymol (v0.99) [55] display of multiple superposed structures obtained with POSA [56].

Structural communication within MCb molecule was investigated using ensembles of native models representing either OO or IO state. These conformational ensembles were represented as networks of interacting amino acids using webPSN [57]. Consensus networks were computed separately for each ensemble to identify possible structural communication signatures. The outputs were also compared to infer commonalities and/or differences between OO state and IO state. The functional significance of the communities identified was investigated by site-directed mutagenesis coupled with CF modeling. Two rounds of mutagenesis were performed: first to identify sites involved in MC conformation switch and provoke conformation exchange and then to probe the impact of candidate communities of interacting residues on this in silico conformation exchange. To this end, the sequence encoding the closest AF2 model to 5M94 (Q5HQ64) was set as starting material to seek mutation combination(s) that could induce modeling of OO states, i.e., mimicking backward switch from IO to OO conformation.

Site-specific evolutionary divergence between MCb and MCg was analyzed using both Diverge 3.0 [58] and Multi-Harmony [59] approaches. Clade-specific consensus sequences at selected sites were displayed using Phylo-mlogo [60]. Molecular genetic evolutionary analysis of MCg1 cluster was initiated using previously characterized sequences [12], plus additional sequences gathered by Psi-Blast searches [61] using either DX4FA5 or A0A149PND7 as queries (cutoff 1e-70). Sequences were aligned using Clustal X [62] and visualized with Phylo-mlogo to derive an MCg1 selective sequence pattern based on the previously established pattern for MCg clade [12]: G[ASP]G[LAVSTIM][LM][VI]AVGY[MIV]DPGNWAT[DEASG]x(30,100)[IVL]A[CT][DA][LV]AE[VIL][IVLA]Gx(5,30)[GAVCSL][TAS][LFYIVC][AVGILST][MLVI]x(50,125)[IVM][LVI]GAT[LVI]MPHN[LI][YF]L[HQ][SGA]x(5,40)[FMLT][LVATIC][VILA]N[SAGL][ASG]x(2,50)[ACS]G[QLM][SN][SA][TA][VLI]T[GAS]. This MCg1-selective pattern used with either DX4FA5 or A0A149PND7 query returned identical PHI-BLAST outputs, which were used as proxy for MCg1 taxonomic distribution. MCg1 seqs were grouped based on pairwise identity before further selection for phylogenetic analysis as previously described [12,63,64,65].

To test the evolutionary scenario that MCg H^+^ network was altered in MCg1 and that further divergence of D4XFA5 cluster tuned MCg1 carrier dynamics, sites that underwent successive changes among MCg clusters were sought to determine whether they would colocalize spatially and demonstrate possible evolutionary coupling, specifically, divergent substitutions of MCg2 residues that correlated first with MCg1 emergence and then with divergence of D4XFA5 cluster. Two test cases were examined: (i) sites showing identical residues in MCg2 and D4XFA5 clusters but distinct in A0A149PND7 cluster and (ii) sites showing cluster-specific residue substitutions.

To increase chances of detecting signals, 3 sets of 11 representative sequences were assembled: one comprising diverse MCg2 seqs [12], another being MCg1 ‘crown’ cluster (A0A149PND7) and the last being *Achromobacter-Bordetella* D4XFA5 MCg1 cluster (Appendix A). Notions supporting this choice were as follows: (i) MCg2 diversity may capture the evolutionary processes operating in this clade as a whole, while (ii) narrow sampling of A0A149PND7 and D4XFA5 MCg1 clusters may indicate discrete divergent steps. The 33 sequences were aligned and subjected as 3 groups of 11 sequences to Multi-Harmony analysis. Sites ranking as the most significant with both Mutli-Relief and Multi-Harmony approaches (exhibiting both z-weight > 6 and z-score < −10, respectively) were mapped on MCg1 models.

## Figures and Tables

**Figure 1 ijms-24-15076-f001:**
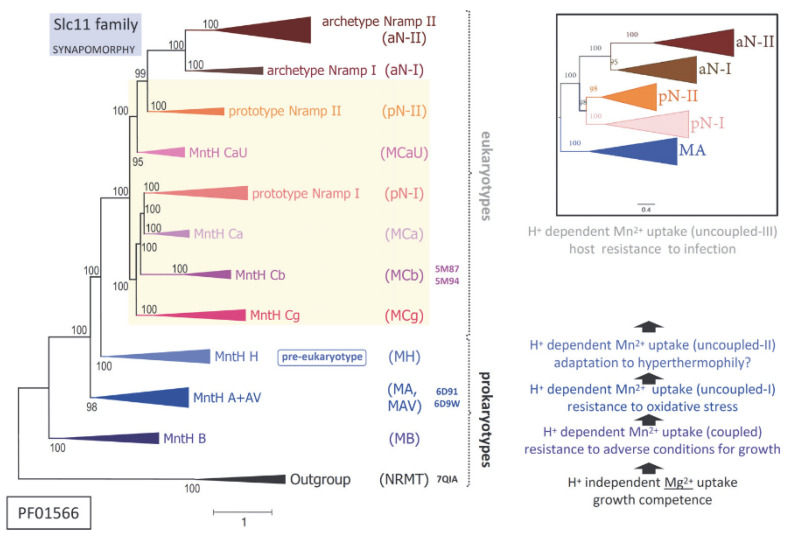
Functional evolution of Slc11 carriers. Schematized Slc11 phylogeny (**Left panel** and Inset) in the context of stepwise evolution of Slc11 H^+^ network interacting with the transmembrane proton motive force to drive Mn^2+^ import (**Right panel**). Note monophyly of bacterial clades MntH Ca, Cb, Cg and eukaryotic Nramp pN-I, as well as established relationships for eukaryotic prototype and archetype *Nramp* parologs (encoding pNs and aNs, respectively, Inset), adapted from [12]. The phylogenetic position of solved structures is color-coded. 7QIA, 6D9W and 5M94 represent carriers inwardly open (IO) while 6D91 and 5M94 exemplify carriers outwardly open (OO). Eukaryogenesis owes at least partly to some Asgard/TACK archaeum intrinsic ability to manage intracellular bacterial symbiont(s) [13]. Based on pre-eukaryotic Nramp synapomorphy, Asgard/TACK MntH H represents an evolutionary intermediate in the transition from bacterial Mn^2+^ permease to eukaryotic antibacterial defense [12]. On the other hand, bacterial MntH Cs inherited Nramp-derived characters distinguishing them from MntH homologs of bacterial origin (e.g., MntH B, MntH A), which perhaps fostered colonization of specific ecological niches [12].

**Figure 2 ijms-24-15076-f002:**
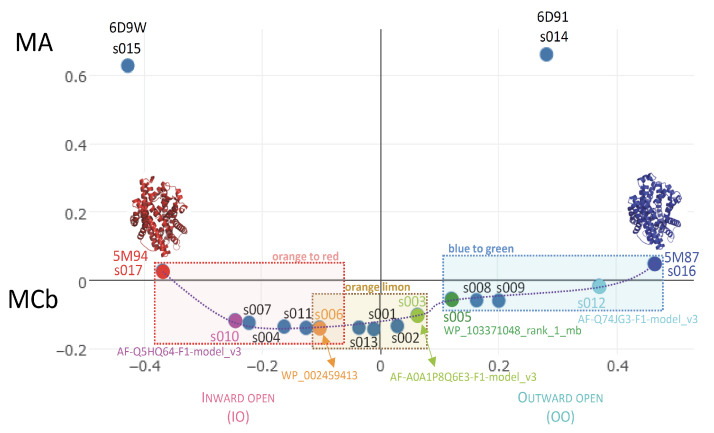
AF2/CF modeling of various MCb predicts a series of possible conformers in carrier cycle. A suite of candidate conformers populating the hypothetical path linking solved 3D structures of MCb in outwardly open (OO, 5M87) and inwardly open (IO, 5M94) states was selected based on Dali ‘all against all’ 3D comparisons using MCb PDB structures as reference points (and equivalent MA structures as outgroup). The 5 selected models segregate with either OO (cyan, green) or IO (magenta, orange) MCb reference structure or neither (limon). They were binned in separate groups (blue-to-green ensemble, limon-orange pair, orange-to-red ensemble) to examine how communities of networked residues evolve during MCb OO-to-IO transition.

**Figure 3 ijms-24-15076-f003:**
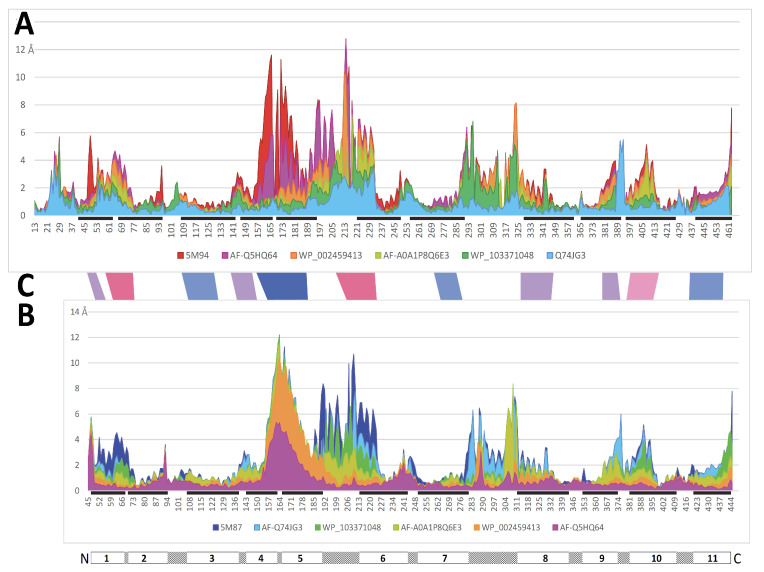
Per-residue RMSD of AF2/CF MCb models relative to both MCb reference structures for each candidate conformer selected. Per-residue Ca RMSD profiles (in Angstroms) were deduced from pairwise structural alignments onto OO (5M87 (**A**)) and IO (5M94 (**B**)) MCb structures, whose respective positions of transmembrane helices 1–11 are indicated below. Progression of local structural changes that occur early (cyan, green) vs. late (orange, magenta) during the OO-to-IO transition suggests tentative ordering of the possible intramolecular rearrangements taking place ((**C**) red to blue).

**Figure 4 ijms-24-15076-f004:**
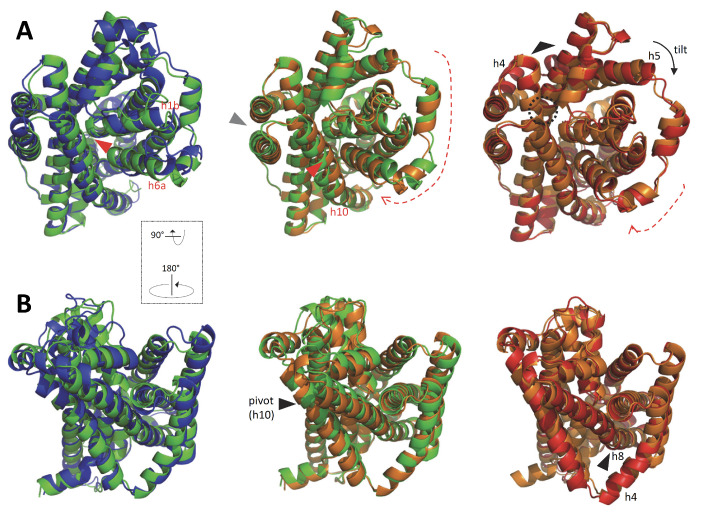
Progression in predicted intramolecular rearrangements during MCb OO-to-IO transition. Multiple superposition (POSA) of two reference 3D structures (OO 5M87 blue and IO 5M94 red) with two CF pdb models (green, orange) shows intramolecular rearrangements that mimic Me^2+^ binding to (Left) and occlusion of (Middle) the external substrate binding site before opening of the inner gate (Right). Alternate views are presented from the cell exterior (**A**) and within the membrane (**B**). Arrowheads and arrows point at areas evolving during MCb transition (see text).

**Figure 5 ijms-24-15076-f005:**
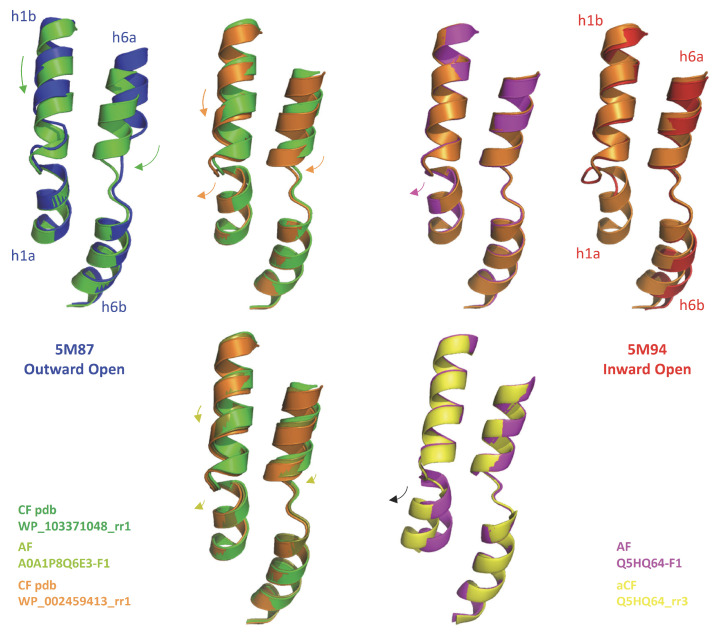
Predicted motion of h1 and h6 during MCb carrier transition from outwardly open to inwardly open state. Motion of helical segments deduced from superposed AF2/CF/aCF models for diverse MCb sequences and solved structures of OO (5M87) and IO (5M94) MCb carriers is indicated with color-coded arrows. The AF2/CF models studied are presented in Figure 2. Using aCF to model Q5HQ64 allows visualizing motion of h1a in IO state (which is absent from 5M94 structure).

**Figure 6 ijms-24-15076-f006:**
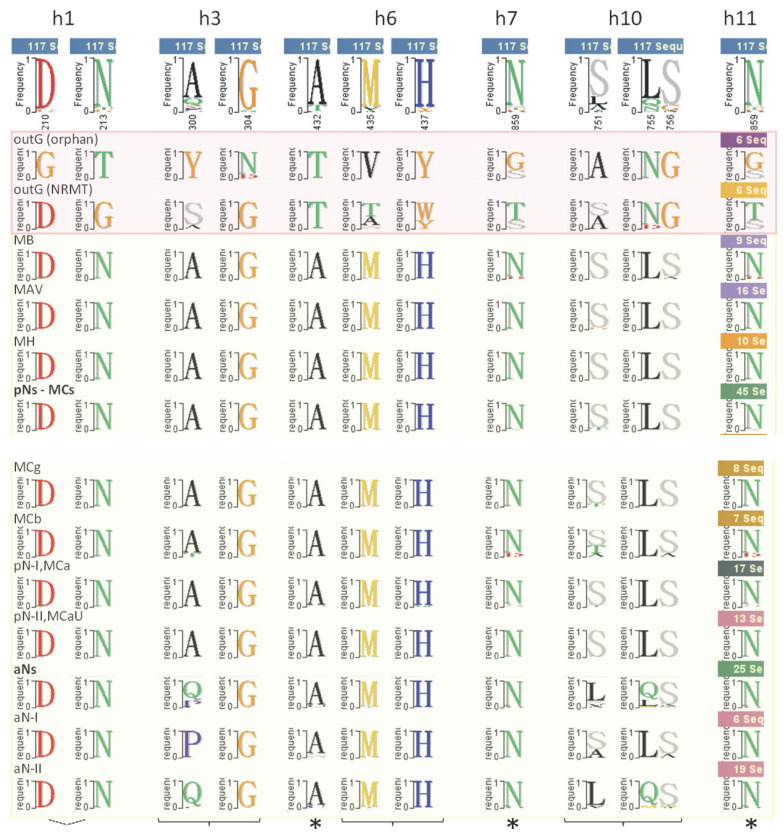
Synapomorphy distinguishing the Slc11 family from phylogenetic outgroups (all part of PFAM01566). The consensus for each clade of the Slc11 family tree (cf Figure 1) is shown for each site representing a type ii evolutionary rate shift between the orphan outgroup and Slc11 carriers. The five combinations of site-directed mutations that were assayed are indicated below: Slc11-specific residues were exchanged for the corresponding outG (orphan/NRMT) residues to study the impact of these evolutionary rate shifts on AF2/CF modeling of Slc11 conformers using both MCb and MCg1 templates. Combinations are indicated with *, braces or a broken line when each mutation was also tested separately.

**Figure 7 ijms-24-15076-f007:**
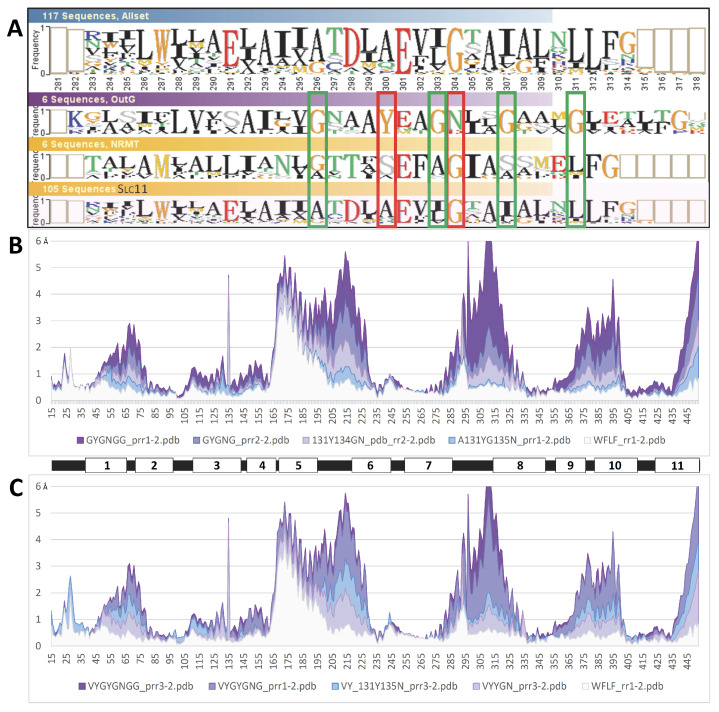
Impact of h3 topology on Q5HQ64 predicted conformation. (**A**). Phylo-mLogo display of the multiple sequence alignment with residues colored according to their chemical character, type ii Slc11/outgroup evolutionary rate shifts indicated in red and outgroup type I evolutionary rate shifts highlighted in green (i.e., 4 invariant Gs in the orphan OutG only). (**B**,**C**) Impact of outgroup h3 type I evolutionary rate shifts on Q5HQ64 conformation in YN (Y-N) background (**B**) and VY YN (h3 Y-N and h6 V-Y) background (**C**) shown as per-residue RMSD (Ca trace) in Å from (wt Q5HQ64 CF pdb model4 used as proxy to) AF2 initial IO model. Q5HQ64 transmembrane helices are positioned below. WFLF, wt Q5HQ64 CF pdb IO conformer (model 2) shows limited opening of l4/5-h5 tilt (inner gate, cf aCF modeling, Figure 5) indicated by substantial local per-residue RMSD. Combining h3 mutations induces synchronous locking of the inner gate and broad-scale rearrangement switching to OO conformation.

**Figure 8 ijms-24-15076-f008:**
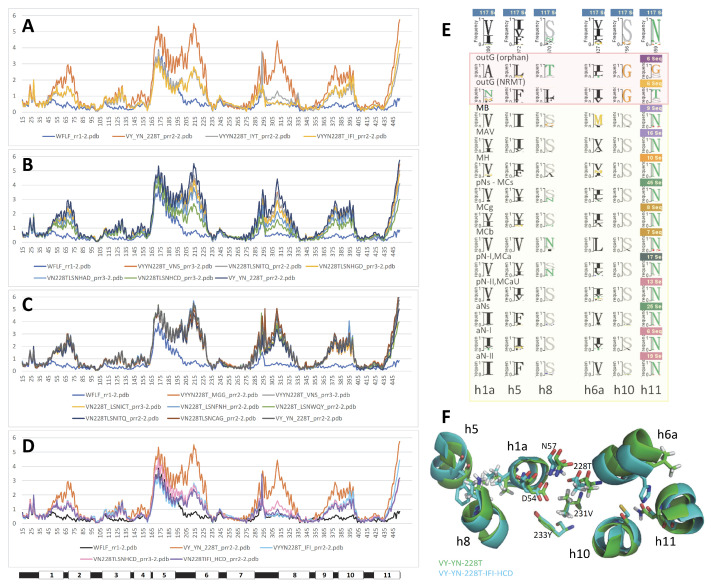
Targeting MCb alternate communities of networked residues prevents modeling Q5HQ64 conformation switch backward from IO to OO state. A-D: Per-residue RMSD of mutant Q5HQ64 (in Å). (**A**) Triple mutation targeting 5M87 community 7 (i.e., Q5HQ64 h1a V50, h5 V177, h8 N334 cf Appendix A). (**B**,**C**) Triple mutation of the 5M94 community 11 [nc18] (i.e., Q5HQ64 h6a L223, h10 S398, h11 N442 cf Appendix A). (**D**) Targeting both 5M87 community 7 (h1a, h5, h8) and 5M94 community 11 [nc18] (h6a, h10, h11) simultaneously. (**E**) Natural evolutionary variation of each of the targeted sites. (**F**) Impact on the Me^2+^ BS arrangement of the mutations tested.

**Figure 9 ijms-24-15076-f009:**
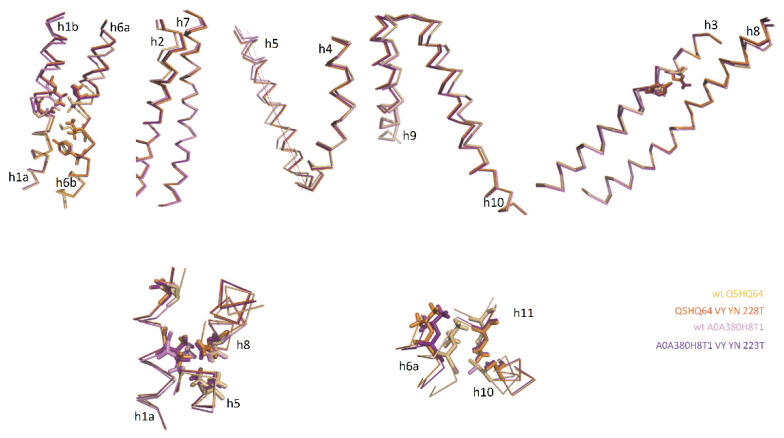
In silico backward conformation switch from IO to OO state is structurally conserved among distinct MCbs. (**Top**) Distinct native MCb IO conformers (Q5HQ64 and A0A380H8T1) and their respective VNT mutation-induced OO models were superposed, and Ca trace deviations are presented separately for either adjacent or linked helices, as indicated. Sticks show residues forming the substrate binding site (h1 and h6) and the conformation-driving mutation h3 YN. (**Bottom**) Details of mutually exclusive communities of networked residues that mediate MCb inner gate opening (**left**) or that lock outer gate (**right**).

**Figure 10 ijms-24-15076-f010:**
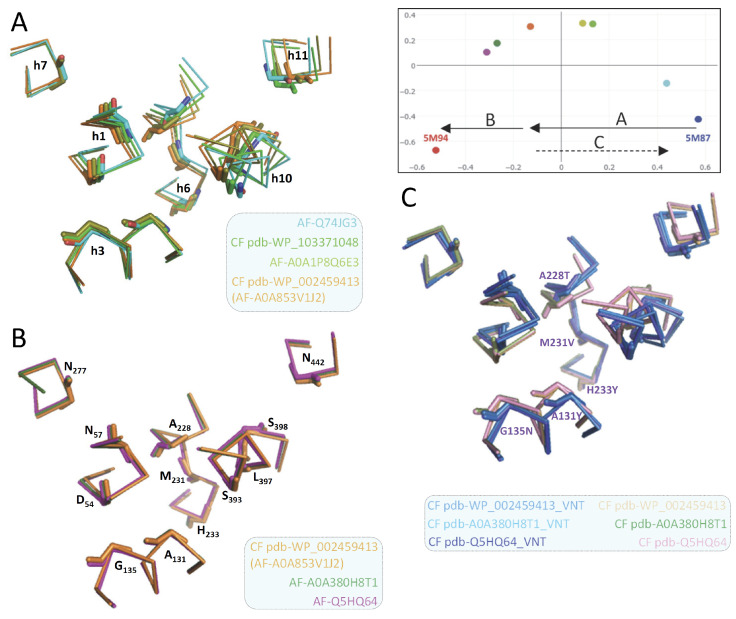
Slc11 synapomorphy articulates MCb model conformation switch mimicked by the mutation VNT. Multiple structural model superpositions display the 3D arrangement of Slc11 synapomorphy, indicated by stick representing residues’ main chains and numbered according to MCb Q5HQ64. For clarity, helix and residue numbers are indicated in separate panels ((**A**,**B**), respectively) and apply to all 3 panels (**A**–**C**). The VNT mutation, i.e., Q5HQ64 h3 A131Y G135N h6 A228T M228V H231Y, is indicated in (**C**). Slc11 synapomorphy 3D rearrangement during native MCb forward conformation switch (**A**), subsequently as MCb carriers open inwardly (**B**) and owing to the VNT mutation (Q5HQ64, A0A380H8T1, WP_002459413 (**C**)). Inset: Correspondence analysis of native MCb models (colored dots as in Figure 2) and delineation of distinct processes: A, forward carrier conformation switch OO to IO and unlocking of the inner gate, and B, locking of the outer gate and opening of the inner gate. Backward conformation switch from IO to OO state that is mimicked by the VNT mutation is also represented (C, cf Appendix A Inset).

**Figure 11 ijms-24-15076-f011:**
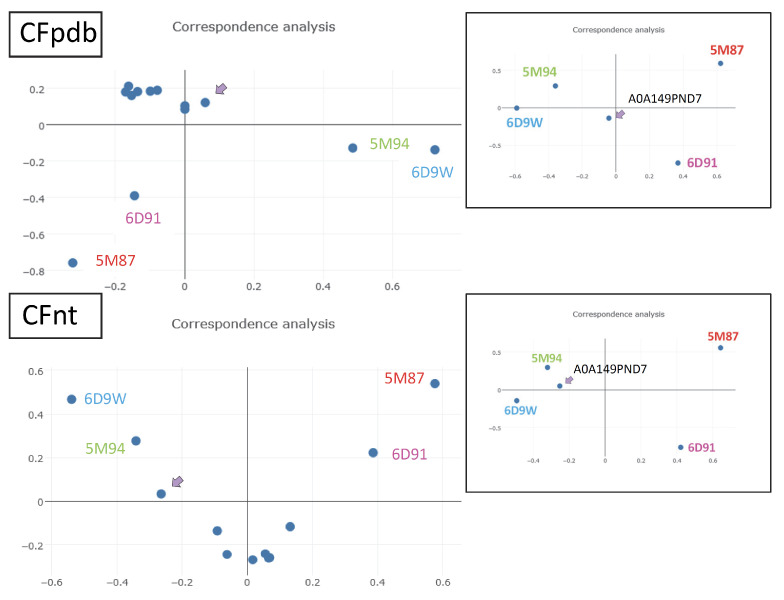
Structural variation among MCg models obtained with either CF with pdb template (CF pdb) or CF with no template (CF nt). (**Left**) Pairwise comparisons (DALI All Against All) of wt MCg models, using MCb reference structures (5M87 and 5M94) and MA (6D91 and 6D9W) structures as outgroup. The unlabeled dots represent the models obtained for select MCg sequences [12]. (**Right**) Re-analyses of the models picked from left (arrow) identify CFnt model of A0A149PND7 as candidate IO conformer.

**Figure 12 ijms-24-15076-f012:**
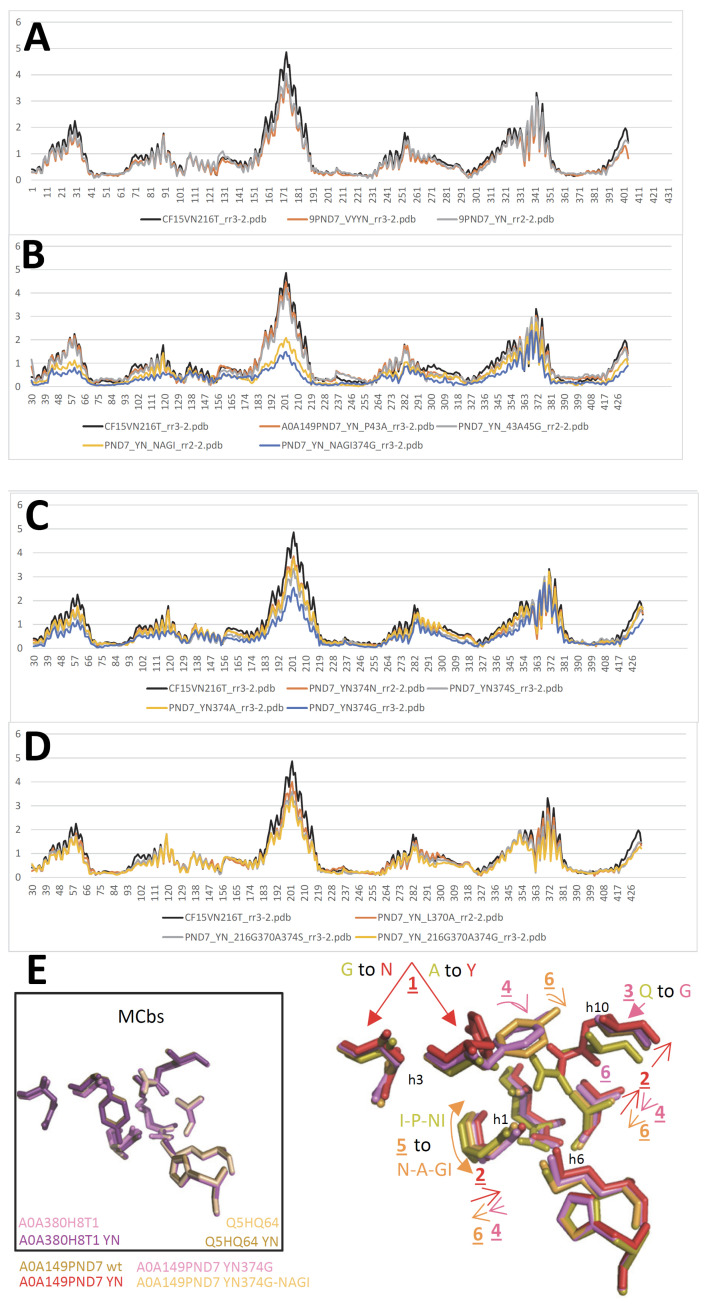
Site-specific suppression of MCg1 A0A149PND7 h3 YN-driven conformation switch. AD. Per-residue RMSD of A0A149PND7 mutants. (**A**) At Slc11-specific sites inducing conformation switch modeling (h3 YN ^+^/_-_ h6 VY and A216T). (**B**) Near h1 Me^2+^ BS and at h10 Q374. (**C**) Alternative substitutions at h10 Q374. (**D**). Combining h3 YN, h10 L370A, h6 A216G and h10 Q374S or G. (**E**) Superposed structures demonstrate mutagenesis (closed arrowheads)-induced deviation (open arrowheads) of select sites in h3, h10, h1 and h6: 1—YN mutation induces 2—Ca deviation of h1, h6 and h10 residues; 3—h10 Q374G mutation allows 4—rearrangement of h3 Y and correction of Ca deviation for h6 and h1 residues; 5—NAGI mutation stimulates 6—further rearrangement of h3 Y and correction of Ca deviation of h6 and h1 residues. Inset: Impact of h3 YN mutagenesis on MCb modeling.

**Figure 13 ijms-24-15076-f013:**
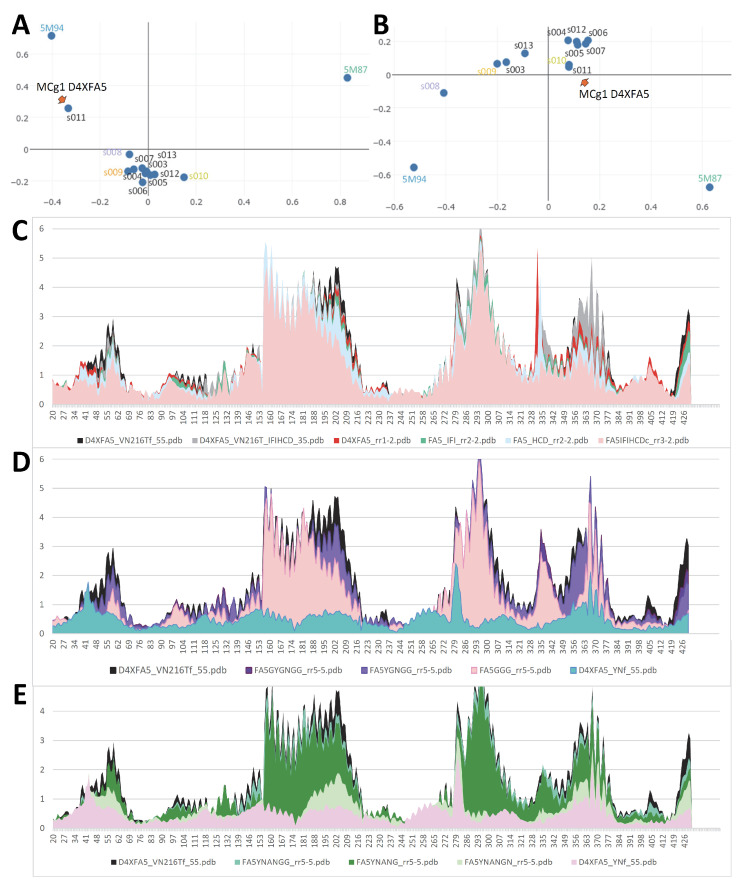
CF nt models candidate alternate conformers of MCg1 D4XFA5. Dali AAA-based re-examination of CF nt models obtained using AF2 training model5 (**A**) or AF2 training model2 (**B**) for MCg homologs (cf Figure 1 or Uniprot proxies) using as reference structures MCg1 A0A149PND7 models (IO, aCF-rr1-2, s008/violet, and its converted OO form, A0A149PND7 VY-YN-216T CF nt-rr3-2, s010/lime) together with the pair of MCb structures OO (5M87) and IO (5M94) used as outgroup. The structure s009/orange is the actual prediction for native A0A149PND7 resulting from each CF nt run. (**C**–**E**) Per-residue RMSD of CF nt models for MCg1 D4XFA5 (wt CF nt rr3-5 used as reference IO model). (**C**) CF nt model rr1-2 is an OO state mimicked by the CF nt model rr5-5 of D4XFA5 VNT mutant (h3 YN h6 VY A216T); it is also partially inhibited by mutations known to impair MCb carrier gating. (**D**) h3 mutagenesis suffices to switch D4XFA5 conformation from IO to OO, but YN mutation is required to move selective segments (h1, h6, h10). (**E**) h11 site D422 modulates cooperation between h3 YN and h10 ANG mutations.

**Figure 14 ijms-24-15076-f014:**
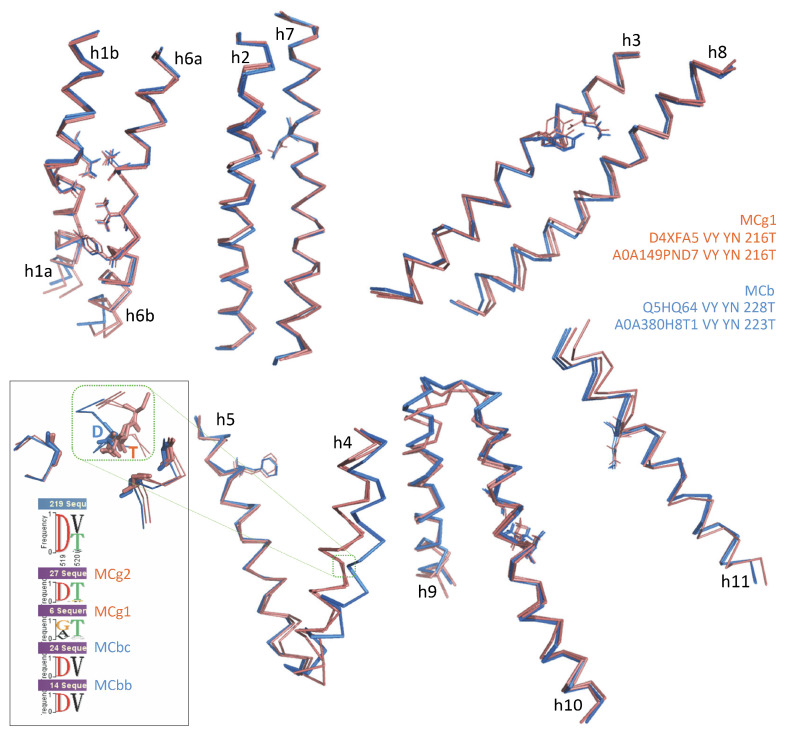
Three-dimensional arrangement of Slc11 synapomorphy is conserved among MCbs and MCg1s. Three-dimensional superposition of VNT mutation-induced OO states for pairs of MCb and MCg1 models. Slc11 synapomorphic sites are highlighted by representing side chains with lines (h1, h3, h6, h7, h10, h11) together with h5 putative pivot point. Inset: Details of MCg1 h4 sequence divergence and 3D rearrangement of the H^+^ network.

**Figure 15 ijms-24-15076-f015:**
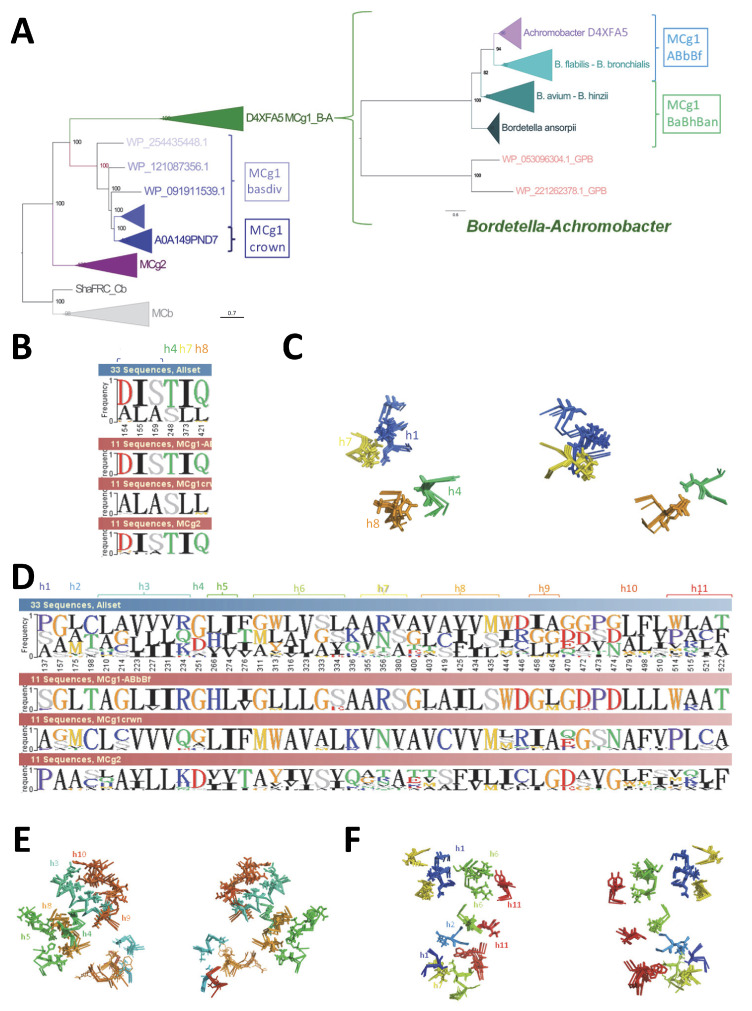
Divergence of the clusters represented by A0A149PND7 (MCg1 crown) and D4XFA5 (MCg1 ABbBf/Bordetella-Achromobacter) shows 3D colocalization of evolutionary coupled mutations. (**A**) Simplified MCg1 phylogeny detailing the 3 groups of sequences that were analyzed (MCg2, MCg1 crown and MCg1 ABbB). (**B**,**D**) Phylo-mlogo displays of two classes of divergent sites that were examined: sites conserved between MCg2 and MCg1 ABbB but distinct in MCg1 crown (**B**) and group-selective sites (**D**). Transmembrane site helix location (h1–h11) is indicated above. (**C**,**E**,**F**) 3D mapping of the divergent sites presented in (**B**,**D**), using the superposed models shown in Figure 14. (**E**) Helices 3, 4, 5, 8, 9 and 10. (**F**) Helices 1, 2, 6, 7 and 11. Two views are presented per panel.

**Figure 16 ijms-24-15076-f016:**
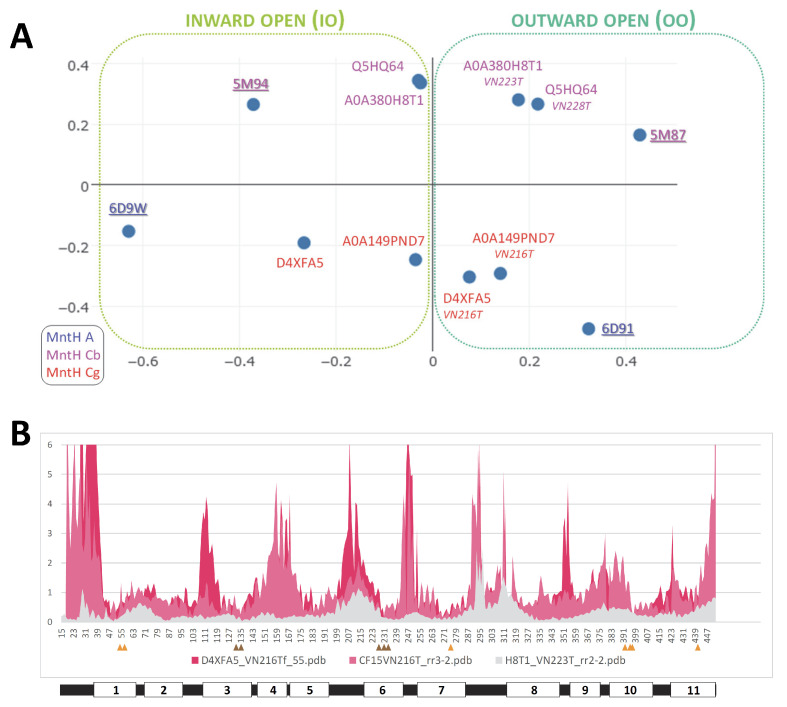
Three-dimensional conservation of Slc11 synapomorphy amid structural divergence of MCb and MCg1 clades. (**A**) Correspondence analysis of Dali all-against-all structural comparisons using pairs of alternate (OO and IO) states for models representing distinct phylogenetic groups (cf Figure 1): the sister groups MntH Cb and MntH Cg1 (eukaryotic origin) and MntH A (prokaryotic origin). PDB references structures underlined. (**B**) Per-residue RMSD (Ca) between MCbs (gray) or between MCg1s and Q5HQ64 (red) OO models (VNT mutants). The position of the sites forming Slc11 synapomorphy is indicated by arrowheads (orange, or brown when mutated) and the location of Q5HQ64 transmembrane helices is depicted below.

**Figure 17 ijms-24-15076-f017:**
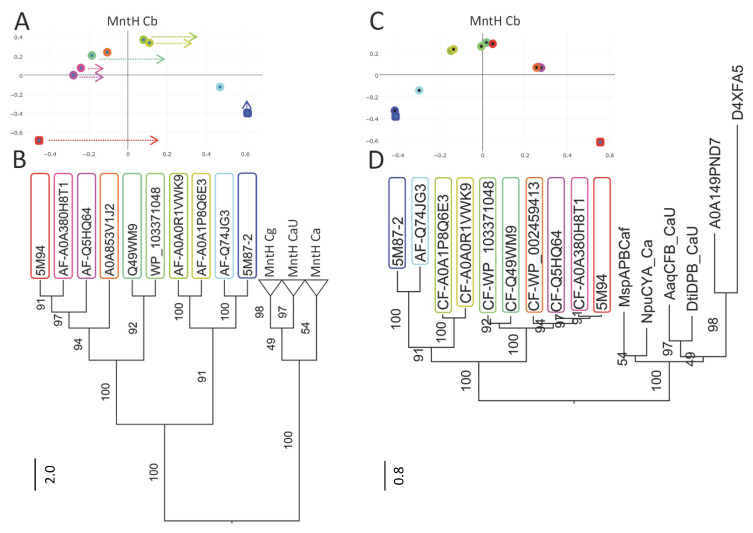
AF2 and CF pdb modeling of MCbs shows strong phylogenetic component. (**A**,**C**) Three-dimensional relationships of models obtained for MCb sequences (dots) using AF2 (**A**) or CF pdb (**C**) and MCb reference structures (squares). Sequence relationships were established with IQ-Tree (306 PI sites) using outgroup sequences from other MC groups. They are represented either as a cladogram (**B**) or a dendrogram showing genetic distances (**D**). MCb sequences are color-coded to highlight four monophyletic clusters. Dotted arrows (**A**) indicate sequences that yield distinct conformers using AF2 or CF pdb modeling.

## Data Availability

All the models used in this study are provided in Appendix A.

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
