# Peer review of "Slc11 Synapomorphy: A Conserved 3D Framework Articulating Carrier Conformation Switch"

_ijms, 2023, doi:10.3390/ijms242015076_

Round 1
Reviewer 1 Report
This interesting research extends the current understanding of the SLC11 family of proton-coupled metal ion transporters. The article is well illustrated with a large number of figures.
Author Response
I thank Reviewer 1 for expressing interest in this work and its presentation and for acknowledging the results extend our understanding of the SLC11 family of proton-coupled metal ion transporters.
Reviewer 2 Report
Dear Author
Thank you for your manuscript submission. Indeed, your work is well-designed with a brilliant presentation.
The Author represents his original in silico article with an interesting topic of "Slc11 synapomorphy: a conserved 3D framework articulating carrier conformation switch"
As the author mentions "the Slc11 family catalyze proton (H+)-dependent uptake of divalent metal-ions (Me2+) such as manganese and iron - vital elements coveted during infection. Slc11 mechanism of high affinity Me2+ cell import is selective and conserved between prokaryotic (MntH) and eukaryotic (Nramp) homologs, though processes coupling the use of the proton motive force to Me2+ uptake evolved repeatedly."
The Author after employing fantastic bioinformatic software tools and databases concludes that: "Prevailing within evolving 3D contexts, Slc11 synapomorphy is thus a reliable phylogenetic marker for function. "
Author Response
Many thanks to Reviewer 2 who enthusiastically judged this work well-designed and brilliantly presented.
Reviewer 3 Report
In this study, the author takes an in silico approach to ask fundamental questions regarding the evolution and structure-function of the extended SlcII family of proton coupled divalent metal transporters. Using state of the art Alphafold and Colabfold 3D predictions, Dr. Cellier was able to devise novel conformational change models for how these transporters act to move a metal across a membrane. The findings will be of interest to the community of researchers investigating this family of proton-metal transporters and also to researchers interested in the application of alphafold and colabold to unravel protein structure-function conundrums. These studies would be strengthened if the authors added a wet bench study to validate some of these interesting models. Nevertheless, the investigation should inspire others in the field to test these predictions.
Suggestions for improvement:
1) The manuscript is written in a way that would be greatly appreciated by in silico structural biologists, but may not be so easily comprehended by the diverse readers of IJMS, including wet bench researchers. Its very jargony and I found it difficult to follow at times.
2) A vast amount of the data is in supplementary materials which makes following the data tedious. Is it possible to bring some of the major points in as Figures?
3) The discussion is too long and should be more concise.
Author Response
I thank reviewer 3 for appraising this work advances knowledge of the evolution of the Slc11 family and of the application alphafold and colabold tools to ask fundamental questions regarding protein structure-function relationships. Indeed, it would be rewarding that some of the predictions presented inspire wet bench validation studies by other investigators.
The suggestions for improvement of the manuscript were followed:
- Text was revised to limit the use of in silico structural biologist jargon,
- 9 Figure supplements were integrated into the body of the article,
- The length of the discussion was reduced.
It is felt that these modifications enhance the presentation of this work.